

# A light-weight $NO_2$ to $NO_x$ conversion model for quantifying $NO_x$ emissions of point sources from $NO_2$ satellite observations

Sandro Meier[1,2], Erik F. M. Koene[1], Maarten Krol[3,4], Dominik Brunner[1], Alexander Damm[2], and Gerrit Kuhlmann[1]

[1]Swiss Federal Laboratories for Materials Science and Technology (Empa), Dübendorf, Switzerland
[2]Remote Sensing Laboratories, Department of Geography, University of Zurich, Switzerland
[3]Meteorology and Air Quality, Wageningen University & Research, Wageningen, The Netherlands
[4]Institute for Marine and Atmospheric Research Utrecht (IMAU), Utrecht University, The Netherlands

**Correspondence:** Sandro Meier (sandro.meier@empa.ch)

**Abstract.**

Nitrogen oxides ($NO_x$ = NO + $NO_2$) are air pollutants which are co-emitted with $CO_2$ during high-temperature combustion processes. Monitoring $NO_x$ emissions is crucial for assessing air quality and for providing proxy estimates of $CO_2$ emissions. Satellite observations, such as those from the TROPOspheric Monitoring Instrument (TROPOMI) on board the Sentinel-5P

satellite, provide global coverage at high temporal resolution. However, satellites measure only $NO_2$, necessitating a conversion to $NO_x$. Previous studies applied a constant $NO_2$-to-$NO_x$ conversion factor. In this paper, we develop a more realistic model for $NO_2$ to $NO_x$ conversion and apply it to TROPOMI data of 2020 and 2021. To achieve this, we analysed plume-resolving simulations from the MicroHH Large Eddy Simulation model with chemistry for the power plants Bełchatów (PL), Jänschwalde (DE), Matimba and Medupi (ZA), as well as a metallurgical plant in Lipetsk (RU). We used the cross-sectional

flux method to calculate NO, $NO_2$ and $NO_x$ line densities from simulated NO and $NO_2$ columns and derived $NO_2$-to-$NO_x$ conversion factors as a function of the time since emission. Since the method of converting $NO_2$ to $NO_x$ presented in this paper assumes steady state conditions as well as that the conversion factors can be modeled by a negative exponential function, we validated the conversion factors using the same MicroHH data. Finally, we applied the derived conversion factors to TROPOMI $NO_2$ observations of the same sources. The validation of the $NO_2$-to-$NO_x$ conversion factors shows that they can account for

the $NO_x$ chemistry in plumes, in particular for the conversion between NO and $NO_2$ near the source and for the chemical loss of $NO_x$ further downstream. When applying these time-since-emission-dependent conversion factors, biases in $NO_x$ emissions estimated from TROPOMI $NO_2$ images are greatly reduced from between –50 and –42% to only –9.5 to –0.5% in comparison with reported emissions. Single-overpass estimates can be quantified with an uncertainty of 20–27%, while annual $NO_x$ emission estimates have uncertainties in the range of 4–21% but are highly dependent on the number of successful retrievals.

Although more simulations covering a wider range of meteorological and trace gas background conditions will be needed to generalize the approach, this study marks an important step towards a global, uniform, high-resolution, and near real-time estimation of $NO_x$ emissions - especially with regard to upcoming $NO_2$ monitoring satellites such as Sentinel-4 and -5 and CO2M.





# 1 Introduction

Nitrogen oxides ($NO_x$ = NO + $NO_2$) are reactive trace gases and important air pollutants since they cause oxidative stress when respired, are involved in the formation of ground-level ozone ($O_3$) and particulate matter and contribute to acid rain (Thurston, 2017). As most $NO_x$ emissions originate from high-temperature combustion processes, monitoring these sources is crucial for air quality regulation and can be used to estimate the (co-emitted) $CO_2$ emissions, provided that one knows the $CO_2$:$NO_x$ emission ratio for a given source (e.g., Goldberg et al., 2019a; Kuhlmann et al., 2021; Liu et al., 2020; Reuter et al.,

2019; Hakkarainen et al., 2023). Estimating $CO_2$ emissions from large sources such as power plants and cities will be an important component of the $CO_2$ Monitoring and Verification Support (CO2MVS) service that is currently being developed under the European Copernicus $CO_2$ project (CoCO2) in support of the Paris Agreement (Pinty et al., 2017; Janssens-Maenhout et al., 2020). For this purpose, emission data should be available in near real time. A convenient method to obtain such high-resolution, uniform global emission estimates is to use satellite observations (Pinty et al., 2017).

Several case studies have investigated the potential and limitations of quantifying point source $CO_2$ emissions from space (e.g., Bovensmann et al., 2010; Goldberg et al., 2019b; Kuhlmann et al., 2021; Nassar et al., 2017; Reuter et al., 2019). One of the methods to quantify emissions is the cross-sectional flux method, which determines emissions by dividing a plume into several cross-sections. By integrating the measured vertical column densities along a cross-section, a line density is obtained. Each line density can be converted into a flux by multiplication with an effective wind speed representing the mean transport

speed of the plume. Under the assumption of steady-state conditions, the flux at each cross-section along the plume can be used to estimate the emissions(Varon et al., 2018).

Estimating $CO_2$ emissions from $NO_2$ satellite data is appealing because $NO_2$ can be measured with higher accuracy than $CO_2$. There are also a number of existing and upcoming satellites that provide $NO_2$ products with high temporal and spatial coverage in comparison with $CO_2$ satellites. The most prominent existing instrument is the TROPOspheric Monitoring Instru-

ment (TROPOMI) on the Sentinel-5 Precursor satellite, which provides daily observations of $NO_2$ and other trace gases with a spatial resolution of $3.5 \times 5.5\,\mathrm{km}^2$ at nadir (van Geffen et al., 2022; Veefkind et al., 2012). Several case studies have shown that TROPOMI data can be used to estimate $NO_x$ emissions from cities and power plants (e.g., Douros et al., 2023; Goldberg et al., 2019b; Lorente et al., 2019).

Satellite-based radiance data only allow for the retrieval of $NO_2$ but not NO. However, more than 90% of $NO_x$ from combus-

tion processes is emitted as NO, which is then partially oxidized to $NO_2$ inside the plume (Pronobis, 2020; Seinfeld and Pandis, 2016). To retrieve $NO_x$ emissions, it is therefore necessary to convert $NO_2$ to $NO_x$. Previous studies often used a constant $NO_2$ to $NO_x$ conversion factor of about 1.32 derived assuming steady-state conditions (e.g., Beirle et al., 2011; de Foy et al., 2015; Kuhlmann et al., 2021; Beirle et al., 2021). Recent studies that used regional chemistry transport model simulations derived conversion factors in the range of 1.1 to 1.9, but acknowledge that the values near sources are likely larger (e.g., Lorente et al.,

2019; Rey-Pommier et al., 2022; Goldberg et al., 2022; Hakkarainen et al., 2024).

In the CoCO2 project, plume-resolving large eddy simulations of atmospheric transport with chemistry were conducted using the MicroHH model (van Heerwaarden et al., 2017). These simulations showed that the $NO_x$:$NO_2$ ratios inside the plume are





highest near the source and decrease roughly exponentially with increasing time after emission (Krol et al., 2023). Figure 1 schematically depicts the evolution of NO, NO$_2$ and NO$_x$ concentrations in a plume. While more than 90% of NO$_x$ is emitted
as NO (Pronobis, 2020), it is rapidly oxidised to NO$_2$ in the presence of ozone (O$_3$), titrating the available O$_3$. Only after dilution and mixing of the plume with ambient air along the plume, the concentration of O$_3$ starts to increase again, leading to the oxidation of further NO. As a result, the ratio of NO$_x$ to NO$_2$ is largest shortly after the emission and gradually decreases over time. The rate of this oxidation process depends on several factors such as the amount of NO$_x$ emitted, the concentration of O$_3$ and volatile organic compounds (VOCs), as well as photolysis rates and meteorological conditions. Subsequently, NO$_2$
is mainly removed by reacting with OH radicals with lifetimes ranging from hours to a few days in the lower troposphere (Seinfeld and Pandis, 2016). According to figure 1, NO$_x$ decays exponentially with a constant e-folding lifetime, but in reality the lifetime may change along the plume due to changing OH radical concentrations.

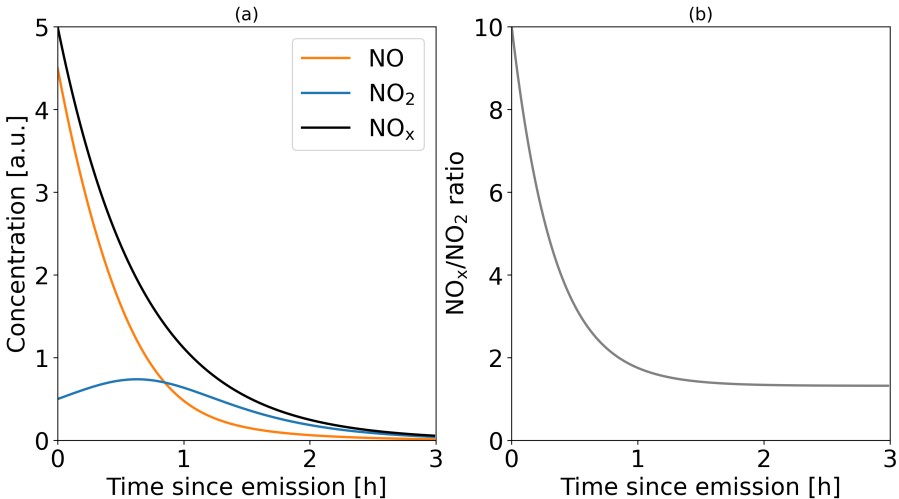

**Figure 1.** (a) Schematic illustration of NO, NO$_2$ and NO$_x$ concentrations after the emission of NO$_x$ where 90 % of NO$_x$ is emitted as NO. (b) Resulting NO$_x$/NO$_2$ ratio.

Since the NO$_x$:NO$_2$ ratio inside plumes cannot be assumed constant, the aim of this study is to develop a more realistic model for a conversion factor of NO$_2$ to NO$_x$ that accounts for the spatiotemporal variations of NO$_x$ chemistry in plumes.
The model is applied in combination with the cross-sectional flux (CSF) method, which were both implemented in the Python package for "data-driven emission quantification" (Kuhlmann et al., 2024, *ddeq*). To develop a more realistic NO$_2$-to-NO$_x$ conversion that varies with time since emission and hence with the distance of the cross-section from the source, we use MicroHH simulations that were conducted within the CoCO2 project. Simulations were performed for the power plants in Bełchatów (PL), Jänschwalde (DE) and Matimba & Medupi (ZA) (hereafter referred to as Matimba), as well as a metallurgical
plant in Lipetsk (RU). The derived parameterisation is then applied to TROPOMI observations of these four sources over a two-year period.



## 2 Data and Methods

### 2.1 Development of a NO$_2$ to NO$_x$ conversion model using MicroHH simulations

#### 2.1.1 Estimating emissions with the cross-sectional flux method

As illustrated in Figure 1, the chemistry of NO$_x$ progresses as a function of time. Therefore, it is necessary to convert the length of a plume into a time since emission. The CSF method implemented in *ddeq* is particularly useful for this purpose, as it divides a plume into several cross-sections perpendicular to the plume direction and establishes a plume-following coordinate system with along-plume and across-plume coordinates.

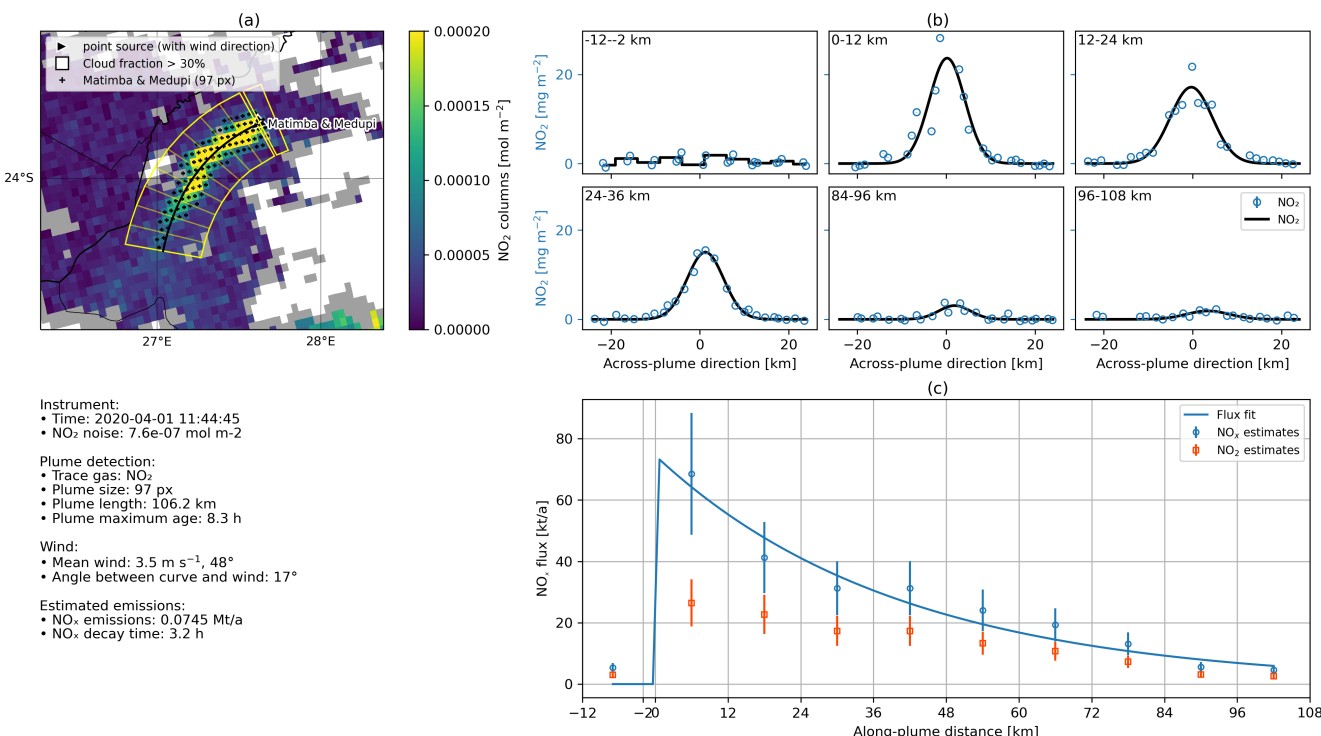

**Figure 2.** Example of the cross-sectional flux method. (a) Satellite image of a NO$_2$ plume divided into sub-polygons. (b) Integrated trace gas concentration for six sub-polygons. (c) Estimated trace gas fluxes as mass NO$_2$ along the plume and the corresponding fitted function to estimate the emissions.

Figure 2 shows the application of the CSF method for a TROPOMI NO$_2$ image containing the plume from the Matimba and

Medupi power plants in South Africa. Note that the power plants are only 6 km apart, so that their plumes appear as a single plume in the TROPOMI image. *ddeq* uses a plume detection algorithm to determine the location of the plume (Kuhlmann et al., 2019). A center line is drawn along the ridge of the plume, which is used to compute along- and across-plume coordinates (denoted by $x$ and $y$, respectively), and to outline the plume area (yellow polygon). The polygon is subdivided in sub-polygons



of 12 km length (Kuhlmann et al., 2020). For each sub-polygon, the mass of the trace gas enhancement over the background
$\Delta\Omega$ $[\mathrm{g\,m^{-2}}]$ is integrated over the width of the plume which yields line densities $q$ $[\mathrm{g\,m^{-1}}]$ at distance $x$:

$$q(x) = \int_{y_{min}}^{y_{max}} \Delta\Omega(x,y)\,dy \tag{1}$$

The plume width is defined as twice the maximum distance of a detected plume pixel from the center of the curve. As an alternative, the line density $q$ can be computed by fitting a Gaussian curve to the enhancements inside the polygon, perpendicular to the direction of the plume:

$$g(y) = \frac{q}{\sqrt{2\pi}\sigma}\exp\left(-\frac{(y-\mu)^2}{2\sigma^2}\right) \tag{2}$$

with $g$ being the fitted column using the standard width $\sigma$ and center position $\mu$ to the observations (Kuhlmann et al., 2021). Figure 2b shows the computation of the line densities for six examples at different distances from the plume. The line densities computed from $NO_2$ observations need to be converted to $NO_x$ line densities using an $NO_2$ to $NO_x$ conversion model $f$ that depends on the time since emission $t$:

$$q_{\mathrm{NO_x}}(t) = f(t) \cdot q_{\mathrm{NO_2}}(t) \tag{3}$$

The time since emission $t$ is computed from an effective wind speed $u_{eff}$ at the source and the arc length of the center line. Details of the estimation of the function $f(t)$ are presented in Section 2.1.3.

Next, the line densities are converted to fluxes $F$ by multiplying them with $u_{eff}$. Finally, the emission $Q$ is estimated by fitting a negative exponential function to the fluxes $F(t)$, the additional fit parameter $\tau$ representing the $NO_x$ lifetime.

$$F(t,\tau) = Q \cdot \exp\left(-\frac{t}{\tau}\right) \tag{4}$$

### 2.1.2 Synthetic Satellite Observations

To gain a better understanding of the $NO_x$:$NO_2$ ratios in plumes, simulations from high-resolution atmospheric transport models incorporating chemistry are needed. For this study, we used the large-eddy simulation (LES) model MicroHH (van Heerwaarden et al., 2017), which was recently extended with a chemistry module (Krol et al., 2023). The model simulated reactive trace gases and $CO_2$ as well as meteorological variables such as temperature, pressure, and wind speed. MicroHH included a simplified version of the chemistry scheme implemented in the IFS model of the European Centre for Medium Range Weather Forecast (Huijnen et al., 2016), simulating the species $O_3$, $NO$, $NO_2$, $NO_3$, $N_2O_5$, $HNO_3$, $CO$, $CO_2$, $CH_4$ (fixed), $H_2$ (fixed), $HO_2$, $OH$, $H_2O_2$, $CH_2O$, $RO_2$, and $ROOH$, as well as $C_3H_6$ as a representative of VOCs. The chemistry was tuned to match the $NO_x$ and $HO_x$ chemistry of IFS and to realistically represent the photostationary state between $NO$, $NO_2$ and $O_3$.

The MicroHH model was run on a $128 \times 128 \times 4$ km domain in the longitude, latitude and altitude directions respectively for Matimba and a $51.2 \times 51.2 \times 4$ km domain for Bełchatów, Jänschwalde, and Lipetsk. The spatial resolution was set to 100



$\times\,100 \times 25$ m for the Matimba case and $50 \times 50 \times 25$ m for the others. Each case was simulated for 48 hours, starting at 00:00 UTC, and the output was saved hourly. The model was initialised and driven with hourly meteorological data from the ERA5 reanalysis. For the background concentrations of trace gases, reanalysis data from the Copernicus Atmosphere Monitoring Service (CAMS) were used (Krol et al., 2023; van Stratum et al., 2023). To simulate the plumes, typical quantities of $NO_x$ emissions from bottom-up reported values of previous years were released at the respective locations of the power plants and industrial facilities. The $NO_x$ emissions were split into 95% NO and 5% $NO_2$ by mass (Krol and van Stratum, 2021).

**Table 1.** Details of the four MicroHH simulations used in this study (Krol et al., 2023)

| Facility | Country | Simulation period |
| --- | --- | --- |
| Power plant Jänschwalde | Germany | 22 - 23 May 2018 |
| Power plant Bełchatów | Poland | 6 - 7 June 2018 |
| Steel plant Lipetsk | Russia | 12 - 13 June 2019 |
| Power plant Matimba and Medupi | South Africa | 24 - 25 July 2020 |

The model output consisted of 3D data of the reactive trace gases as well as meteorological variables such as temperature, pressure, and wind speed. The output was post-processed into 2D datasets resembling synthetic satellite observations but without including any measurement noise. The resolution was degraded to the expected resolution of the CO2M satellites of $2 \times 2$ km. For the wind speeds, a 2D weighted average of the 3D wind fields was calculated based on the vertical emission profile. These wind speeds are used to estimate the simulated emissions. The specific model settings and boundary conditions used for the MicroHH model runs are described in Krol and van Stratum (2021) and Krol et al. (2023) while the post-processing is documented in Koene and Brunner (2023).

### 2.1.3 Conversion of $NO_2$ to $NO_x$ line densities in MicroHH

To derive a more realistic conversion model of $NO_2$ to $NO_x$ line densities $f(t)$, the vertically integrated MicroHH simulations were analysed for the sources Bełchatów, Jänschwalde, Lipetsk, and Matimba by applying the CSF method as outlined above.

We analysed the time steps 8 to 14 UTC for both simulated days instead of only the ones at TROPOMI overpass time to derive more robust $NO_2$-to-$NO_x$ conversion factors that better represent varying atmospheric and site conditions. For each polygon of the detected plumes, the line densities of NO and $NO_2$ were calculated. The along-plume distance of each plume was divided by the profile-weighted wind speed at the source to convert them to a time since emission. Such a conversion allows us to account for the effects of varying wind speeds on the concentration of trace gases. For each source, we fitted a negative exponential function to the median $NO_x$:$NO_2$ ratio, using the standard deviations of the analysed time steps as uncertainties.

$$f(t) = m \cdot \exp\left(-r \cdot t\right) + f_0 \tag{5}$$

The fitting parameter $m$ represents a scaling factor, $r$ the rate at which the $NO_x$:$NO_2$ ratio decreases and $f_0$ the offset to which the ratio will converge to with time. The resulting conversion factor $f(t)$ can be multiplied with the corresponding $q_{NO_2}(t)$





line densities to obtain $q_{\mathrm{NO_x}}(t)$. The uncertainty $\sigma_f$ of $f$ is calculated from the fitted uncertainties of the three parameters by propagation of uncertainty

$$\sigma_f(t) = \sqrt{\left(\frac{\partial f(t)}{\partial m}\right)^2 \sigma_m^2 + \left(\frac{\partial f(t)}{\partial r}\right)^2 \sigma_\tau^2 + \left(\frac{\partial f(t)}{\partial f_0}\right)^2 \sigma_{f_0}^2} \tag{6}$$

and used to update the uncertainty $\sigma_l$ of the $\mathrm{NO_x}$ line densities $q$:

$$\sigma_q = \sqrt{f^2 \sigma_q^2 + \sigma_f^2 q^2} \tag{7}$$

The method of converting $\mathrm{NO_2}$ to $\mathrm{NO_x}$ presented in this paper relies on the assumption of steady state conditions as well as an exponential decay of the conversion factor (Eq. 5). Therefore, it is important to check if $\mathrm{NO_x}$ emission estimates derived from the time-dependent algorithm are consistent with the emitted quantities. For this purpose, we estimated the $\mathrm{NO_x}$ emissions of the same daytime time steps of MicroHH three times: once using the modelled $\mathrm{NO_x}$ fields, once using the $\mathrm{NO_2}$ fields and applying the constant $\mathrm{NO_2}$-to-$\mathrm{NO_x}$ conversion factor of 1.32 (referred to as the algorithm with constant factor), and once using the negative exponential function fitted above as conversion factors (referred to as the time-dependent algorithm).

## 2.2 Application to TROPOMI $\mathrm{NO_2}$ satellite observations

### 2.2.1 TROPOMI Satellite Observations

We applied the method of converting $\mathrm{NO_2}$ to $\mathrm{NO_x}$ line densities developed in the current study to the latest processing version (v2.4.0) of the tropospheric $\mathrm{NO_2}$ observations from TROPOMI for the years 2020 and 2021. In accordance with van Geffen et al. (2019), only data with quality assurance values higher than 0.75 were utilised. In addition, we downloaded the auxiliary data comprising 3D $\mathrm{NO_2}$ fields from the 3D chemistry-transport model TM5-MP to recompute the air mass factors (AMFs) (see Section 2.2.2) (Eskes and van Geffen, 2021)). We set the precision of the retrieved tropospheric $\mathrm{NO_2}$ vertical column densities (VCDs) to $7.6 \cdot 10^{-7}\ \mathrm{kg\,m^{-2}}$, which corresponds to $1 \cdot 10^{-15}\ \mathrm{molecules\,cm^{-2}}$. This is an average uncertainty over polluted regions and corresponds to approximately 20% of the measured $\mathrm{NO_2}$ VCDs (van Geffen et al., 2019). Using a constant precision is needed because the precision of tropospheric VCDs in the TROPOMI product is correlated with the total $\mathrm{NO_2}$ VCDs which causes problems when calculating line densities.

Emissions were estimated by applying the constant and time-dependent algorithms to the AMF-corrected images. For each source, the respective fitting parameters $m$, $r$ and $f_0$ from the MicroHH simulations were used to convert $\mathrm{NO_2}$ into $\mathrm{NO_x}$. Estimates were then aggregated by month and annual emissions were estimated as the median of the monthly statistics. This was done to avoid a potential bias due to an unbalanced number of data points per month. For estimating the uncertainty of the annual emissions, a seasonal cycle was fitted to all emission estimates using a cubic Hermite spline with periodic boundary conditions (Kuhlmann et al., 2021). The corresponding uncertainty $\sigma_e$ accounts for the uncertainties of the single-overpass estimates through error propagation. To further account for uncertainties in the diurnal ($\sigma_d$) and seasonal ($\sigma_s$) cycles, the total uncertainty $\sigma_{\mathrm{tot}}$ was calculated as follows:

$$\sigma_{\mathrm{tot}} = \sqrt{\sigma_e^2 + \frac{\sigma_d^2}{n} + \frac{\sigma_s^2}{n}} \tag{8}$$



Here, both $\sigma_d$ and $\sigma_s$ were set to 30% according to Hill and Nassar (2019). As the estimated $NO_x$ emissions with the time-dependent algorithm depend on the $NO_2$-to-$NO_x$ conversion factor, a sensitivity analysis was performed by applying the $NO_2$-to-$NO_x$ conversion factors calculated for Jänschwalde and Matimba to all four sources.

### 2.2.2 Air mass factor correction

For the retrieval of $NO_2$ VCDs, a priori $NO_2$ profiles from a 3D chemistry transport simulation called TM5-MP are used. Due to its coarse resolution of $1° \times 1°$, the model cannot resolve individual plumes but rather represents them as smeared out $NO_2$ enhancements. Consequently, the TM5-MP pixels have neither the correct concentration profile of the plume nor the correct background concentration, but a mixture of both. This tends to lead to an overestimation of AMFs and consequently an underestimation of VCDs within the observed plumes and, vice versa, outside of the plumes. Such a bias over polluted regions is known from previous studies (Griffin et al., 2019; Verhoelst et al., 2021; Douros et al., 2023). To address these biases we constructed a more realistic $NO_2$ profile that is representative for the observed plumes. To this end, we interpolated the auxiliary data from the TM5-MP model and the ERA5 planetary boundary layer (PBL) height data to the higher resolution TROPOMI pixels. We set the $NO_2$ mole fraction within the PBL to $5 \cdot 10^{-9}$ mol/mol for all detected plume pixels of the images for the years 2020 and 2021. This is an average $NO_2$ concentration within the PBL of detected plumes based on the four MicroHH simulations, independent of the along-plume distance. However, we acknowledge that the profile concentration should ideally decrease along the plume. With the new $NO_2$ profiles $x_{new}$, we recalculated the AMFs according to Eskes et al. (2022):

$$\text{AMF}_{new}(x_{new}) = \text{AMF}_{old}(x_{old}) \cdot \frac{\sum_l A_l \cdot x_{new,l}}{\sum_l x_{new,l}}. \tag{9}$$

Finally, we updated the VCDs inside the detected plumes for all images using the recalculated $\text{AMF}_{new}$:

$$\text{VCD}_{new} = \frac{\text{VCD}_{old} \cdot \text{AMF}_{old}}{\text{AMF}_{new}} \tag{10}$$

We only recalculated the AMFs and VCDs of detected plume pixels because no other anthropogenic sources other than the power plant or steel plant in focus were simulated by MicroHH. Thus, the $NO_x$ concentrations were too low to obtain representative background concentrations.

### 2.2.3 ERA5 wind data

The CSF method requires wind data to convert trace gas line densities into fluxes. For this purpose, we weighted the 3D wind fields of the ERA5 reanalysis ((Hersbach et al., 2018)) with a profile representing the expected vertical distribution of emissions for power plants in Brunner et al. (2019) and integrated them vertically to obtain 2D wind fields. As in Kuhlmann et al. (2021), we assumed a fixed wind speed uncertainty of 1 m/s for the error propagation in the emission estimation.



### 2.2.4 Comparison with bottom-up reported $NO_x$ emissions at high temporal resolution

Since the year 2000, member states of the European Union have been required to report the emissions of air and water pollutants from large point sources (European Parliament and the Council of the European Union). These data were made publicly

available in 2006 through the European Pollutant Release and Transfer Register (E-PRTR). The database contains the annual emissions of pollutants from nine major sectors such as energy production or metal processing and is available on the European Industrial Emissions Portal (https://industry.eea.europa.eu/). We use the bottom-up reported emissions to assess the accuracy of our emission estimates. We obtained data as annual $NO_x$ emissions from the Jänschwalde power plant for the years 2020 to 2021. For the Bełchatów power plant, the data are only available up to 2017. Therefore, we used the $CO_2$ and $NO_x$ emissions

for 2017 to extrapolate the expected emissions for the years 2020 to 2021 according to Nassar et al. (2022). For the metallurgical plant in Lipetsk, no accurate data on emissions were available since there are no bottom-up reported emissions for this specific site in the annual report of the operating company NLMK. Moreover, from the reports it is not clear if emissions from the captive power plants at the Lipetsk site are included in the reported emissions. For the Matimba and Medupi power plants, monthly emissions are provided by the operating company Eskom.

215  For all three power plants, we interpolated the annual and monthly bottom-up reported $CO_2$ and $NO_x$ emissions to hourly and daily temporal resolution by weighting them with the power plant's energy output according to Nassar et al. (2022). For the European power plants, we used the hourly electricity generation from the transparency platform of the European Network of Transmission System Operators for Electricity (ENTSO-E) (https://transparency.entsoe.eu/). For the Matimba and Medupi power plants, we used the daily electricity production provided by the operating company Eskom.

## 3 Results

### 3.1 $NO_x$ to $NO_2$ ratios in plumes

An example of the $NO_2$ and $NO_x$ vertical column fields simulated with MicroHH for the Matimba case as well as the corresponding $NO_x$:$NO_2$ ratios is depicted in Figure 3. The spatiotemporal patterns of $NO_x$:$NO_2$ ratios of all simulations are displayed in Figure A1.

225  The evolution of $NO_x$:$NO_2$ ratios in the MicroHH model as a function of time since emission is summarized in Figure 4 for all four cases Bełchatów, Jänschwalde, Lipetsk, and Matimba. The figure contains results from all hourly time steps between 8 and 14 UTC from both simulated days.

 Panel (a) shows the median and standard deviation of the ratios, while (b) depicts the corresponding fitted negative exponential functions and the fitted standard deviations. The figure confirms our expectation that the $NO_x$:$NO_2$ ratios are largest close

to the source and decrease with increasing distance downwind. The ratios are generally much larger than the previously used conversion factor $f_0$=1.32 (black horizontal line) and only approach this value at distances larger than 50-100 km and only in some cases. Because most of the $NO_x$ is emitted as NO, NO concentrations close to the source are very high, which leads to complete titration of $O_3$ present in background air and therefore limits the production of $NO_2$ through the oxidation of NO





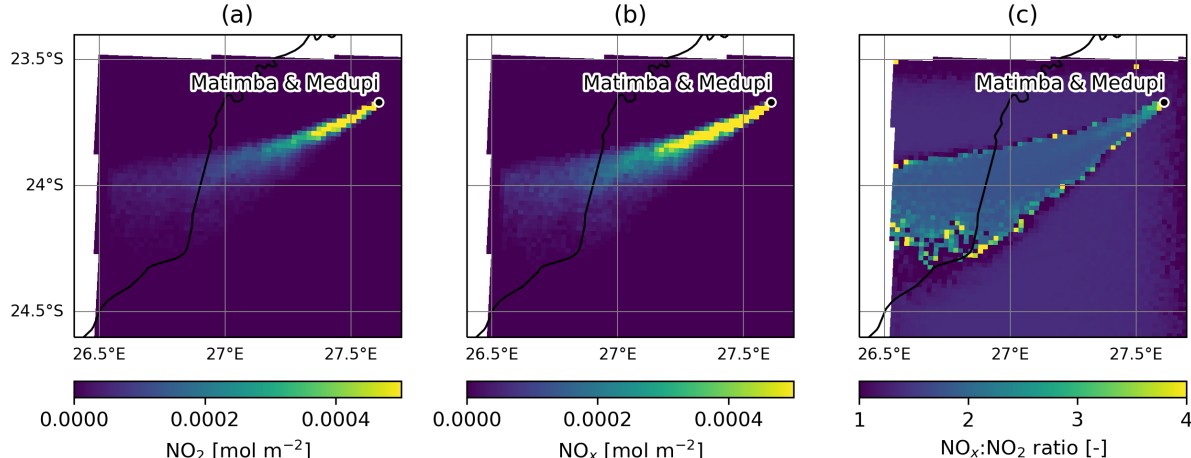

**Figure 3.** Simulated $NO_2$ (a) and $NO_x$ (b) fields as well as the resulting $NO_x$:$NO_2$ ratios (c) from time step 32 (08:00 UTC) of the MicroHH simulation of Matimba.

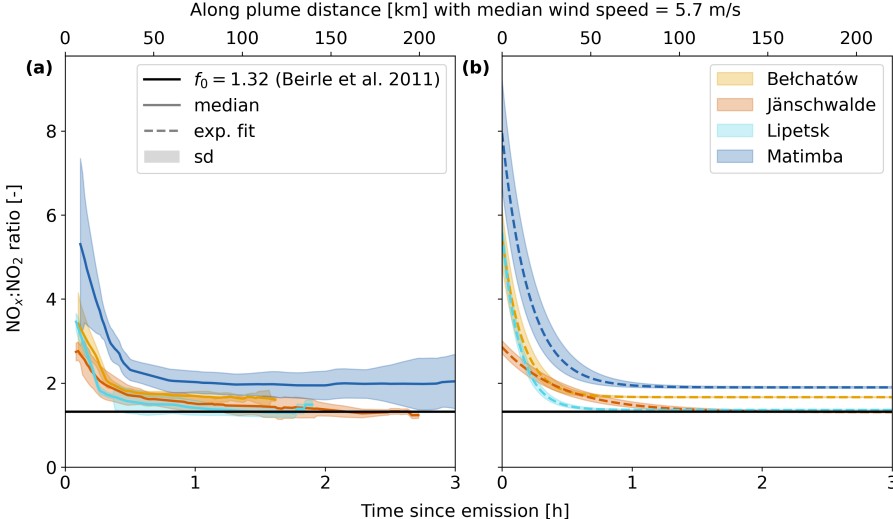

**Figure 4.** Mean $NO_x$ : $NO_2$ ratios of the MicroHH time steps 8 – 14 UTC as a function of time since emission. (a) Median and standard deviation. (b) Fitted negative exponential function and corresponding standard deviation. The time axis is converted to a space axis using the median wind speed in all analysed plumes.

by $O_3$. With increasing dilution and mixing of the plume with background air downwind of the source, the concentration of

NO decreases while the concentration of $O_3$ increases. This accelerates the oxidation of NO and gradually shifts the photostationary equilibrium ratio of NO:$NO_2$ towards $NO_2$ and reduces the $NO_x$:$NO_2$ ratio accordingly (Seinfeld and Pandis, 2016). Compared to the other three simulations, the $NO_x$:$NO_2$ ratio of the Matimba simulation is higher both at the source and further



downwind. The main reason for this behaviour is the amount of $NO_x$ emitted: the more that is emitted, the longer it takes for the plume to mix in sufficient $O_3$ from the surrounding air masses to reach the background photostationary state for $NO_x$ (Krol
et al., 2023). Another reason for the different $NO_x$:$NO_2$ ratios are the meteorological conditions which determine how fast the plumes are mixed with surrounding air masses. Furthermore, the solar irradiances and background concentrations of $O_3$ and VOCs have an strong influence on the $NO_x$:$NO_2$ ratios (Seinfeld and Pandis, 2016). In all four simulations, the ratios level off half an hour after the emission or 50 km along the plume, assuming a median wind speed of the analysed time steps of about 5.7 m/s. Furthermore, Figure 4 illustrates that the standard deviation close to the source is largest for Bełchatów, which leads to
a higher uncertainty in the fitted function. The corresponding fitting parameters of the $NO_x$:$NO_2$ ratios and their uncertainties are listed in Table 2.

**Table 2.** Fitting parameters of the negative exponential function in Eq. 5 to the mean $NO_x$:$NO_2$ ratios of the four MicroHH simulations for the steps 8 to 14 UTC.

| Source | $m$ [$-$] | $\frac{1}{r}$ [min] | $f_0$ [$-$] |
|---|---|---|---|
| Bełchatów | $3.8 \pm 0.7$ | $9.1 \pm 0.8$ | $1.66 \pm 0.01$ |
| Jänschwalde | $1.6 \pm 0.1$ | $27.3 \pm 2.7$ | $1.31 \pm 0.01$ |
| Lipetsk | $4.2 \pm 0.3$ | $8.1 \pm 0.4$ | $1.36 \pm 0.02$ |
| Matimba | $6.1 \pm 1.3$ | $12.4 \pm 1.4$ | $1.90 \pm 0.02$ |

To convert $NO_2$ line densities into $NO_x$ line densities, they are multiplied with $f(t)$ following Eq. (5). The results for the Matimba plume are shown in Figure 5. As a result of the multiplication, the $NO_x$ line densities peak at the source and approximately follow an exponential decay similar to the schematic in Figure 1. In contrast, the $NO_2$ line densities peak
between 20 to 30 km.

Applying the $NO_2$-to-$NO_x$ conversion factors to the MicroHH data as a validation in Figure 6 shows that the estimated $NO_x$ emissions with the time-dependent algorithm are in good agreement with the estimates from the modelled $NO_x$ fields. However, the spread of the estimates is larger when converting $NO_2$ to $NO_x$, which is due to the assumption that the conversion can be modelled by a negative exponential function. Furthermore, emissions estimated from the modelled $NO_x$ fields should align
with the prescribed emissions. However, the emissions are overestimated for Bełchatów and Jänschwalde and underestimated for Matimba, which is due to uncertainties of the CSF method (see Section 4).

Similar to the emission estimates, the estimated $NO_x$ decay times using the time-dependent algorithm are more consistent with those from the modelled $NO_x$ fields whereas the estimates using the algorithm with constant factor are more than twice as high. This overestimation with the algorithm with constant factor is due to the fact that $NO_2$ decreases less rapidly than $NO_x$
due to the gradual shift in the NO:$NO_2$ photostationary equilibrium ratio towards $NO_2$ as mentioned earlier.

The improved agreement between the estimates from modelled $NO_x$ fields and the time-dependent algorithm shows that this model of converting $NO_2$ to $NO_x$ accounts for the $NO_x$ chemistry in the plumes simulated by MicroHH quite well. The larger discrepancies for Bełchatów and Jänschwalde compared to the other cases are probably due to the higher relative



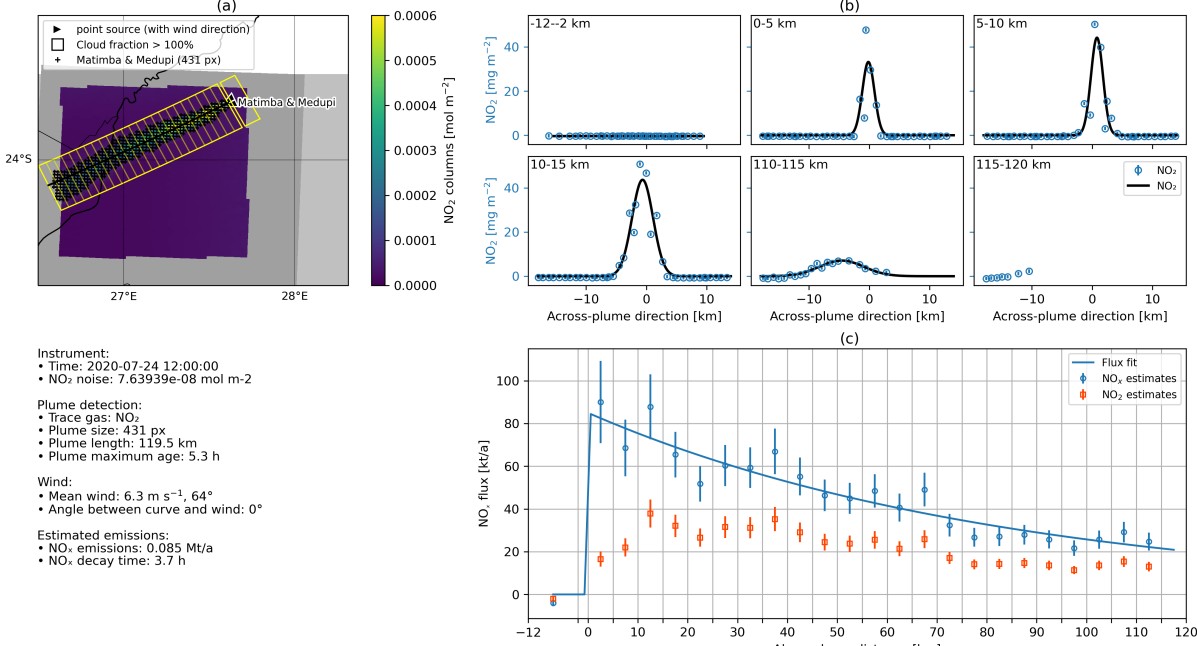

**Figure 5.** Example of estimating $NO_x$ emissions from Matimba MicroHH simulation using the time-dependent $NO_2$-to-$NO_x$ conversion for the cross-sectional flux method implemented in *ddeq*.

uncertainties of the fitted $NO_2$:$NO_x$ ratios as seen in Figure 4b. Nevertheless, the estimated emissions and lifetimes are a
significant improvement over the approach of converting $NO_2$ to $NO_x$ using a constant factor of 1.32.

### 3.2    Application of the $NO_2$-to-$NO_x$ conversion to TROPOMI observations

For the years 2020 and 2021, a total of 737 TROPOMI images were available for Bełchatów, 807 for Jänschwalde, 862 for Lipetsk and 454 for Matimba. However, for the first three sources, only about 7% of the images were sufficiently cloud-free. For Bełchatów and Jänschwalde, the plume detection only worked for half of these cloud-free images due to the proximity
to other coal-fired powerplants. As a result, the plumes often mixed, rendering the estimation of emissions impossible. For Matimba, almost half of the total available images were cloud-free, with plume detection working on more than 80% of these images due to the remote location. An example image of the emission estimation for TROPOMI can be seen in Figure 2.

The AMFs computed for the four cases result in a mean increase of the VCDs inside the plume by a factor of $1.11 - 1.35$ (see Figure A2). The estimated $NO_x$ emissions from the AMF-corrected TROPOMI data for the years 2020 and 2021 are presented
in Figure 7 and listed in Table 3. While the emissions estimated with the algorithm with constant factor only amount to 48–69% of the bottom-up reported emissions, the emissions derived with the time-dependent algorithm are more in line and reach about 88–109%. For all four sources, these estimates are within one standard deviation of the bottom-up reported emissions.



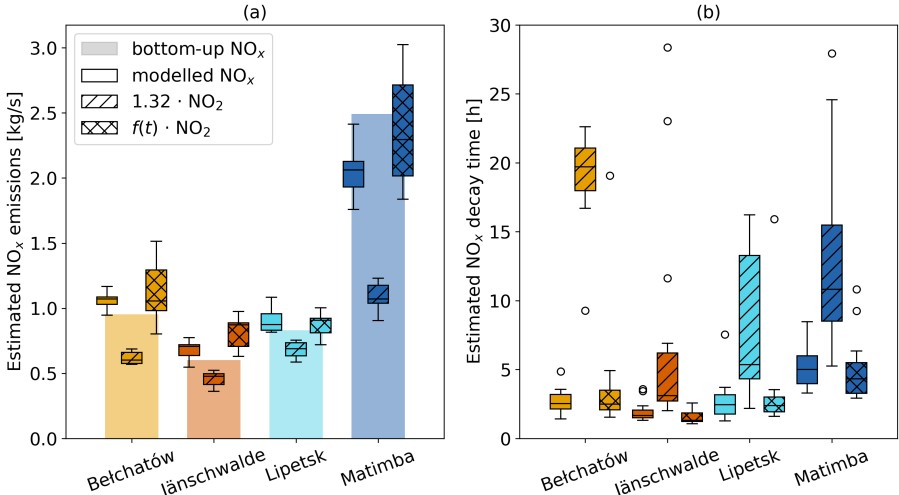

**Figure 6.** (a) Comparison of estimated $NO_x$ emissions against the prescribed (bottom-up) emissions and (b) estimated $NO_x$ decay times using the constant and time-dependent algorithms as well as the modelled $NO_x$ fields. Only the daytime time steps of the MicroHH simulations were utilised. Boxes in the histograms represent the interquartile range (IQR, $25^{th}$ to $75^{th}$ percentile), whiskers the range between $Q1 - 1.5 \cdot IQR$ and $Q3 + 1.5 \cdot IQR$ and circles all data points that fall outside of this range.

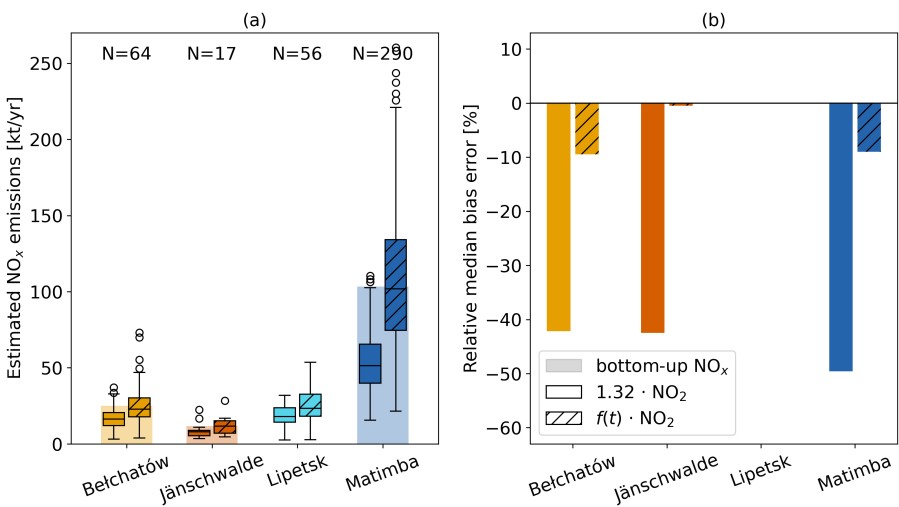

**Figure 7.** (a) Estimated $NO_x$ emissions and (b) their relative median bias errors for Bełchatów, Jänschwalde, Lipetsk, and Matimba for TROPOMI data of the years 2020 and 2021.

Figure 7 also shows that the range of estimated emissions is largest for Matimba with a large number of outliers. The most likely explanation is that the plumes are the longest for this source, meaning that parts of the plume are several hours old and
have likely been subject to different chemistry and wind speeds (see Fig. A6). This leads to strong violations of the assumed



steady-state conditions along the plume and results in outliers in the $NO_x$ fluxes along the plume. The relative mean bias error of a given method is in a similar range for all four sources. While the bias is around –50% relative to the bottom-up reported emissions with the algorithm with constant factor, it is reduced to only –9.5 to –0.5% with the time-dependent algorithm.

**Table 3.** Median and standard deviation of estimated $NO_x$ emissions in kt $NO_2$/a for the years 2020 and 2021 for Bełchatów, Jänschwalde, Lipetsk, and Matimba derived from TROPOMI images

| Source | $1.32 \cdot NO_2$ | $f(t) \cdot NO_2$ | Bottom-up |
|---|---|---|---|
| Bełchatów | $14.6 \pm 4.2$ | $22.0 \pm 8.2$ | 25.1 |
| Jänschwalde | $8.0 \pm 0.7$ | $12.7 \pm 1.9$ | 11.6 |
| Lipetsk | $18.4 \pm 4.1$ | $23.4 \pm 6.8$ | – |
| Matimba | $49.2 \pm 16.6$ | $104.2 \pm 45.8$ | 103.4 |

The uncertainties of the single-overpass and annual estimates are listed in Table 4. The first column shows the median uncertainty of all single-overpass estimates. The second column represents the standard deviation of the difference between estimated and bottom-up emissions. The uncertainties of the first column would agree with those in the second if the bottom-up reported emissions corresponded to the true emissions and all uncertainties were included in the emission estimation. However, the larger magnitude of the values in the second column indicates uncertainties in bottom-up reported emissions (e.g. due to the temporal interpolation) and the presence of other uncertainties in the emission estimation which were not considered. These include the simplified representation of instrument noise, wind speed, and AMF correction. On top of these random errors, there are uncertainties due to systematic errors such as the estimation of background concentrations, the application of the $NO_2$-to-$NO_x$ conversion factors to annual data, and methodological uncertainties, which are not represented in the estimated uncertainties.

The third column in Table 4 shows the uncertainties in annual emissions according to error propagation, while the fourth column additionally accounts for uncertainties in diurnal and seasonal cycles.

**Table 4.** Uncertainties of $NO_x$ emission estimates for single-overpass and annual estimates for Bełchatów, Jänschwalde, Lipetsk, and Matimba

| | Single-overpass estimates [%] | | Annual estimates [%] | |
|---|---|---|---|---|
| Source | Median uncertainty | SD of bottom-up - estimated $NO_x$ | Spline uncertainty | Total uncertainty |
| Bełchatów | 23.2 | 41.7 | 5.9 | 9.5 |
| Jänschwalde | 24.0 | 4.6 | 14.4 | 20.5 |
| Lipetsk | 20.5 | – | 6.9 | 10.6 |
| Matimba | 25.8 | 46.3 | 1.6 | 3.9 |





A simple sensitivity analysis in which the $NO_2$-to-$NO_x$ conversion factors of Jänschwalde and Matimba were applied to all four sources resulted in emission estimates ranging from 10% lower to 50% higher than the estimates shown in Fig. 7. This illustrates that the parameterisation of the $NO_2$-to-$NO_x$ conversion depends on the specific situation such as meteorological conditions, background concentrations, and emission strength, and not representing this situation appropriately adds a signif-
icant uncertainty to the emission estimates. To get a better understanding of these uncertainties, it would be necessary to run more high-resolution chemistry transport simulations covering a wider range of conditions, and to account for these conditions in an extended parameterisation. Despite its simplicity, the first-order parameterisation proposed here, which builds on a small set of high-resolution MicroHH simulations for each source, already leads to a substantial reduction of the bias.

## 4 Discussion

### 4.1 Strengths and weaknesses of the time-dependent $NO_2$-to-$NO_x$ conversion

The analysis of $NO_x$:$NO_2$ ratios in modelled plumes demonstrated the importance of chemical processes leading to a general decrease of this ratio with distance from the source but it also revealed considerable variability from case to case, which is likely the result of different amounts of emitted quantities, photolysis rates, temperatures, wind and turbulent mixing conditions, and trace gas background concentrations. The aim of the time-dependent algorithm developed in this study is to reproduce the $NO_2$
to $NO_x$ conversion of line densities along the plume (Figure 1). If the true chemistry is well approximated, this should lead to a good agreement with the prescribed emissions. The remaining discrepancy is therefore due to deviations from our simplified assumptions (e.g. the assumption of an exponential decay of the ratios along the plume) and due to uncertainties in the CSF method. One important source of uncertainty in the CSF method is the wind speed used to convert line densities to fluxes, which is discussed in section 4.3. The errors of the time-dependent algorithm are more in line with those of the modelled
$NO_x$ fields but are slightly larger because the implemented conversion of $NO_2$ to $NO_x$ does not take into account the specific meteorological and background conditions of each time step, but is based on the median conditions. Thus, the bias is likely to increase when the $NO_2$-to-$NO_x$ conversion factors derived in this study are applied to annual data, as the chemical and meteorological conditions vary considerably during a year. Nevertheless, we argue that applying the four fitted $NO_2$-to-$NO_x$ conversion functions to annual TROPOMI images yields suitable emission estimates because most of the images that can be
used for plume detection were acquired between April and October (see Figure A3). Images taken during the rest of the year often cannot be used for $NO_x$ estimation due to high cloud cover. Consequently, the prevailing conditions for most of the emission estimates are comparable to the conditions in the MicroHH model simulations, which represent days in May to July. As the newly implemented conversion function for $NO_2$ to $NO_x$ showed a significant improvement in the estimation of $NO_x$ emissions and lifetimes from MicroHH simulations, we consider them suitable for the application to TROPOMI images.





## 4.2 Quantification of NO$_x$ emissions using TROPOMI observations

The application of the time-dependent NO$_2$-to-NO$_x$ conversion to the TROPOMI data in Figure 7 has shown that the NO$_x$ emission estimates obtained with the time-dependent algorithm are much closer to the bottom-up reported emissions than the estimates from the algorithm with a constant NO$_2$-to-NO$_x$ conversion factor of 1.32. The relative median bias is reduced from between –50 and –42% to only between –9.5 to –0.5%. However, the significant variance in estimated emissions for Matimba indicates the necessity for further refinement of the approach. One improvement would be to investigate very long plumes that have been subject to different meteorological conditions than those under which the NO$_2$-to-NO$_x$ conversion factors were derived.

As the number of successful emission estimates per year has a strong influence on the uncertainties of the annual emission estimates, maximising the number of suitable satellite images is crucial. Nevertheless, only a fraction of the TROPOMI images could be used for Bełchatów, Jänschwalde, and Lipetsk due to cloud cover. Especially between October and February, emissions could only be estimated for a few days (see Figure A3). The strong seasonal bias in the number of successful estimates may lead to an underestimation of annual emissions as emissions in winter are expected to be larger due to the higher demand of electricity and heating. This gap cannot be filled by the upcoming polar-orbiting Sentinel-5 satellite either but could be alleviated by existing and upcoming geostationary satellites such as GEMS, TEMPO, and Sentinel-4: The hourly temporal resolution increases the probability of obtaining a usable image on a cloudy day. Multiple images during a day would also allow to resolve the diurnal cycle of NO$_x$ emissions, which currently cannot be captured with only one or two overpasses around noon. However, GEMS, Sentinel-4, and -5 have a coarser resolution compared to Sentinel-5P. The complications caused by a coarse spatial resolution can be seen in the example of Jänschwalde: As there are two coal-fired power plants in the vicinity of Jänschwalde (e.g., the Boxberg and Schwarze Pumpe power plants), the plumes often mix, which is why the emissions cannot be estimated reliably using the CSF method. This applies to a lesser extend to Bełchatów. In contrast, fewer sources that could lead to overlapping plumes are located around Lipetsk and Matimba. As shown in Kuhlmann et al. (2021), a satellite with higher spatial resolution, such as CO2M, can help to better differentiate between plumes, mitigating the challenge of overlapping plumes.

The comparison of the uncertainties of the NO$_x$ emission estimates in this study with those in Kuhlmann et al. (2021) highlights the importance of the number of successful emission estimates. The uncertainties of the annual emissions of 4 to 21% in this study are significantly lower than the uncertainties of 16 to 73% and 13 to 52% for two and three of the CO2M satellites in Kuhlmann et al. (2021). The reasons are the higher temporal resolution of TROPOMI compared to CO2M and the high source strength of the power plants considered in the current study. The single-overpass estimates due to random error, in contrast, are only marginally lower in this study than the 29% derived in Kuhlmann et al. (2021). This difference may be attributed to the consideration of additional uncertainties in their study by including a source strength dependent factor and an offset.

The systematic biases due to the application of the NO$_2$-to-NO$_x$ conversion factors to annual TROPOMI data were investigated in the form of a sensitivity analysis. Applying the NO$_2$-to-NO$_x$ conversion factors of Jänschwalde and Matimba to all





four sources resulted in emission estimates ranging from 10% lower to 50% higher, which illustrates that the parameterisation

of the $NO_2$-to-$NO_x$ conversion still adds a significant but unknown uncertainty to the emission estimates. This is because it is not possible to determine how well the conditions under which these parameterisations were derived match those of a given TROPOMI image. However, since most of the suitable satellite images are from the season for which the MicroHH simulations were run, we argue that the calculated $NO_2$-to-$NO_x$ conversion factors are likely to be in good agreement with the conditions of the TROPOMI images.

Overall, the application of the newly developed $NO_2$-to-$NO_x$ conversion factors resulted in more accurate emission estimates compared to the previous constant conversion factor of 1.32. Nevertheless, extrapolating the conversion factors for different meteorological and background conditions remains a challenge.

### 4.3 Effective wind speeds in plumes

Apart from the $NO_x$ chemistry, a realistic representation of the effective wind speed at which the plume is transported is a key

issue. This includes the vertical averaging of 3D wind fields and the consideration of time-varying wind fields. To address the first challenge, the 3D wind speeds were weighted with the expected emission profiles. An advantage of this method is that the weighted winds correspond better to the plume when it is not yet well-mixed within the PBL, i.e. close to the source or in a stably stratified atmosphere. However, with increasing distance from the source, the trace gases become progressively more well-mixed within the PBL. Depending on meteorological conditions, homogeneous mixing can occur within the first few

kilometres of the plume (Krol et al., 2023). In such cases, it would be more reasonable to use the mean wind speed within the PBL. Furthermore, Brunner et al. (2019) have shown that plumes typically rise to a height of 250 m in winter, but up to 360 m in summer. Winds are strongly influenced by the dynamics of the PBL, which has a distinct diurnal cycle, especially in summer. These results suggest that a fixed emission profile is likely not sufficient to vertically weigh the 3D wind fields. Instead, the effective wind should be calculated dynamically and account for parameters such as stack height, flue gas properties, and

meteorological conditions (Brunner et al., 2019). On the bottom line, further studies are needed to assess the suitability of this method to vertically average the wind speeds under different conditions.

### 4.4 Impact of air mass factors

The coarse resolution of the a priori $NO_2$ profiles used for the retrieval of $NO_2$ VCDs leads to an underestimation of VCDs within the plume and an overestimation outside. As the $NO_2$ background VCDs are subtracted from the plume enhancements,

updating the $NO_2$ profiles both within and outside the plume would further increase $NO_x$ emission estimates. This would lead to higher emission estimates and possibly an overestimation which would be in line with the slight overestimation of $NO_x$ emissions when using the time-dependent algorithm in Figure 6 for the same reasons as discussed in section 3.1.

Ideally, the a priori $NO_2$ profiles of the TM5-MP model should be replaced by profiles from higher resolution models such as GEM-MACH (Goldberg et al., 2019b) or CAMS-regional (Douros et al., 2023). However, updating the AMFs for all pixels

was beyond the scope of this study. For this reason, the a priori $NO_2$ profiles of plume pixels were replaced by a constant $NO_2$ mole fraction of $5 \cdot 10^{-9}$ mol/mol within the PBL. This resulted in lower AMFs and consequently higher VCDs by a



factor of 1.15 to 1.35. Other studies have calculated significantly larger corrections. For example, Beirle et al. (2019) found that VCDs need to be corrected by a factor of 1.35 for South Africa and 1.98 for Germany. The higher values are attributed to the assumption made by Beirle et al. (2019) that the entire plume is confined between 60 and 200 m above ground level,

where the height-resolved AMFs are typically smaller than at higher altitudes. In contrast, the correction factors in this study were calculated assuming a homogeneous distribution within the PBL, which is more realistic and in line with the MicroHH simulations. Douros et al. (2023) analysed the impact of replacing the TROPOMI a priori $NO_2$ profiles over Europe with data from the higher resolution CAMS-regional model at a resolution of $0.1 \times 0.1°$. They found that the $NO_2$ VCDs increased by a factor of 1.05 for less polluted sites and up to 1.3 for more polluted sites which is in good agreement with the increases in

VCDs calculated in this study.

### 4.5 Bottom-up reported emissions

In this study, knowledge of bottom-up reported $NO_x$ and $CO_2$ emissions is important for two reasons. Firstly, they are used to evaluate the accuracy of the estimated $NO_x$ emissions from satellites. Secondly, reported emissions can be used to convert the estimated $NO_x$ emissions into $CO_2$. For both applications it is crucial to have information on the reliability and accuracy of the

bottom-up reported emissions. However, many of the bottom-up reported $CO_2$ emissions are estimated from fuel consumption, making assumptions about combustion efficiency, fuel purity and other factors, introducing many uncertainties which are difficult to quantify (IPCC, 2006). It is assumed that bottom-up uncertainties for $CO_2$ are in the range of $\pm$ 10% (Gurney et al., 2016) but significantly higher for $NO_x$ (e.g., Zhao et al., 2011). Deviations between estimated and bottom-up reported emissions are therefore not necessarily due to errors in the estimates but could also originate from inaccuracies in the reported

emissions.

## 5 Conclusions

In this study, we derived a more realistic model for $NO_2$-to-$NO_x$ conversion in plumes of large $NO_x$ sources. We derived parameters for this model using high-resolution chemistry transport model simulations. The conversion model was then applied to TROPOMI observations from 2020 and 2021.

The results show that annual $NO_x$ emissions can be reliably estimated with TROPOMI: the discrepancies between bottom-up and top-down estimates were reduced from between –50 and –42% to only between –9.5 to –0.5% with uncertainties ranging from 4 to 21%. These more accurate $NO_x$ emission estimates are important for air quality monitoring and can be used to convert $NO_x$ to $CO_2$ emissions using $CO_2$:$NO_x$ emission ratios, allowing the use of $NO_2$ imaging satellites such as GEMS, TEMPO, Sentinel-4, and -5 to estimate $CO_2$ emission with high temporal resolution. Furthermore, geostationary satellites will

allow to better resolve the diurnal cycle of emissions and could help to reduce a potential seasonal bias by reducing the number of failed emission estimates caused by cloud cover.

This study also highlights several shortcomings of the current approach. More comprehensive and systematic studies are necessary to determine the dependence of the $NO_2$-to-$NO_x$ conversion factor on prevailing conditions such as solar radiation,



temperature, and background concentrations of reactive trace gases. An alternative approach for converting $NO_2$ into $NO_x$ line
densities would be to use machine learning such as neural networks. Once trained with a high-resolution model with chem-
istry like MicroHH, the network could predict the $NO_2$-to-$NO_x$ conversion factors without the need to run a high-resolution
chemistry transport model for each plume. However, a large number of simulations covering a wide range of conditions would
need to be run for proper training and validation of a machine learning model. Furthermore, we mentioned that more research
is necessary to determine how wind speeds should be vertically averaged in plumes and how systematic uncertainties due to
AMFs can be best accounted for.

The time-dependent $NO_2$-to-$NO_x$ conversion model has been implemented in *ddeq* and can be adjusted for different sources
and conditions. An example Jupyter notebook using Python provides easy access to the implementations, enabling users to
estimate $NO_x$ emissions from $NO_2$ satellite observations of specific sources using their own set of $NO_2$-to-$NO_x$ conversion
parameters. These emissions can then be converted to $CO_2$ emissions using $CO_2$:$NO_x$ ratios. Therefore, the current study is
an important step towards global, uniform, high-resolution, and near real-time estimation of $NO_x$ and $CO_2$ emissions with the
use of satellites, which is crucial for air quality monitoring and greenhouse gas emission monitoring and verification.

*Code and data availability.* The *ddeq* version 1.0 used for this study is available available on Gitlab.com (https://gitlab.com/empa503/
remote-sensing/ddeq). MicroHH data can be downloaded on Zenodo (Koene and Brunner, 2022). An example notebook on how to use
the $NO_2$-to-$NO_x$ conversion covered in this paper can be found in the supplement.



**Appendix A: Additional figures**

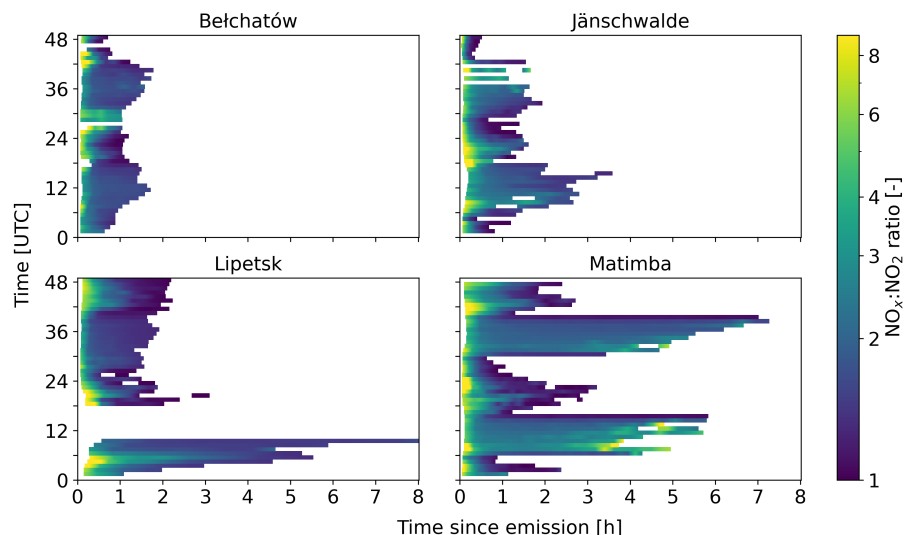

**Figure A1.** $NO_x$:$NO_2$ ratios for 48 individual hourly time steps of the MicroHH simulations of Bełchatów, Jänschwalde, Lipetsk, and Matimba as a function of time since emission, highlighting the spatiotemporal patterns of the $NO_x$ chemistry.

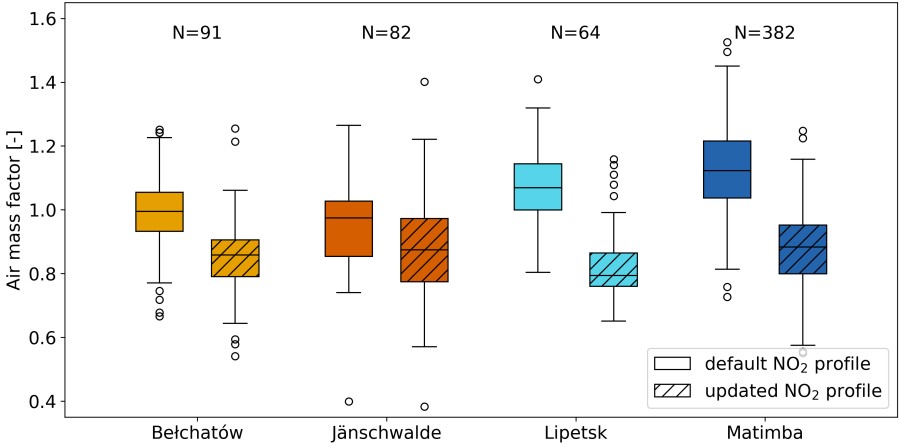

**Figure A2.** Default and updated AMF of TROPOMI images of Bełchatów, Jänschwalde, Lipetsk, and Matimba for the years 2020 and 2021. For the updated AMFs, the $NO_2$ mole fraction was set to $5 \cdot 10^{-9}$ mol/mol within the PBL of the detected plumes



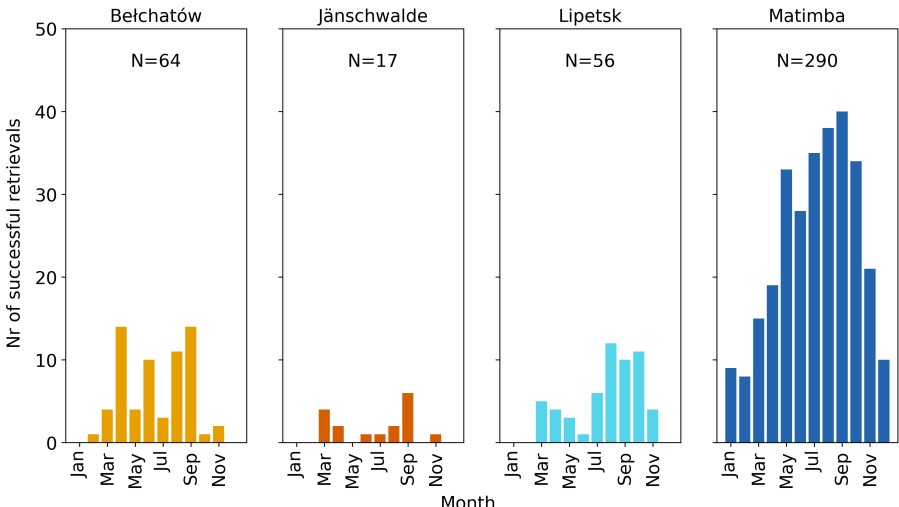

**Figure A3.** Number of successful NO$_x$ emission estimates per month using TROPOMI for 2020 and 2021



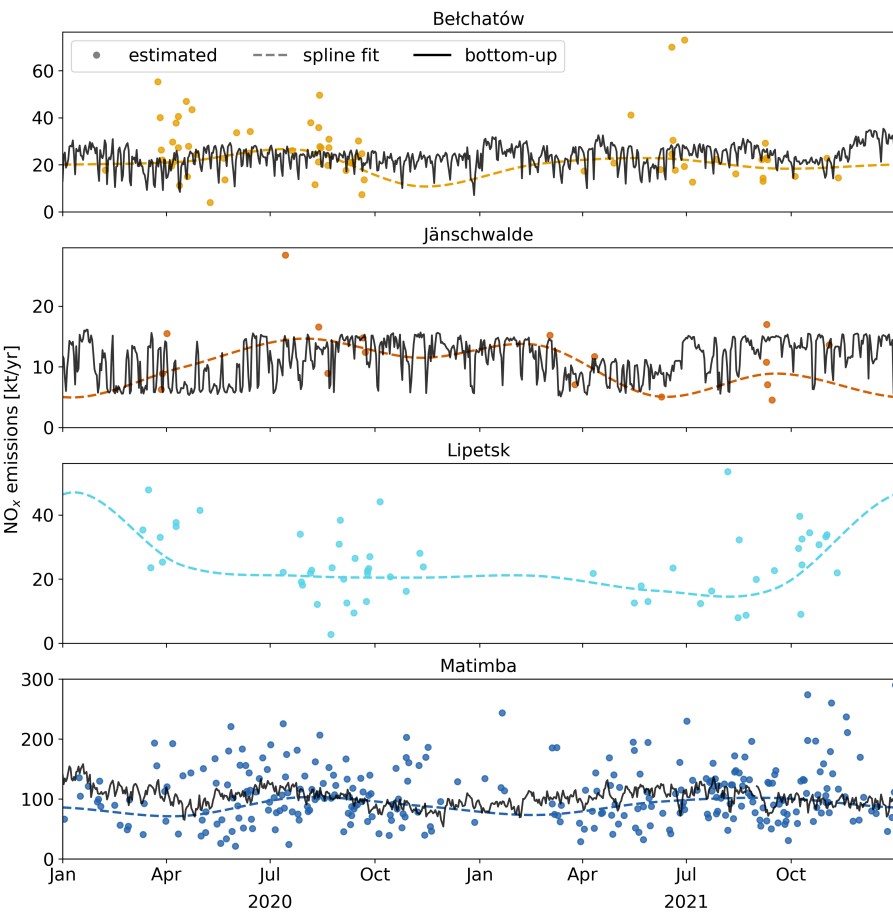

**Figure A4.** Time series of NO$_x$ emission estimates using TROPOMI and bottom-up reported emissions for the years 2020 and 2021. To each time series, a cubic Hermite spline with periodic boundary conditions was fitted.



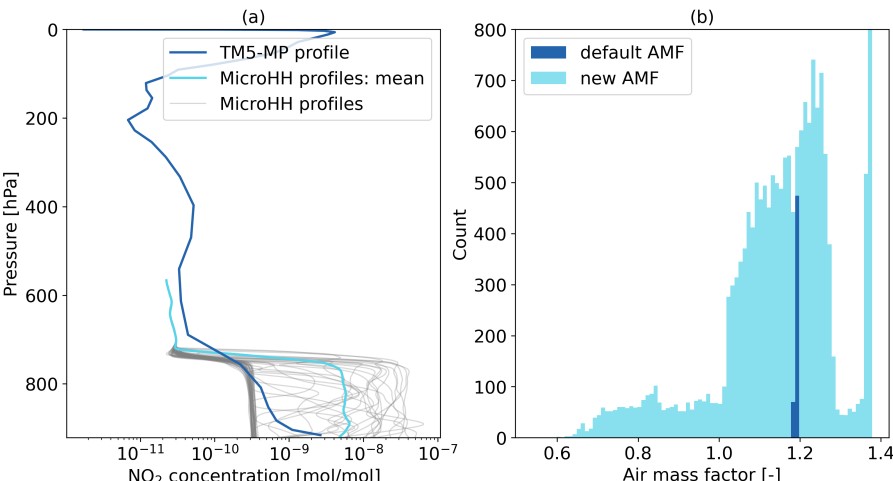

**Figure A5.** (a) TM5-MP and MicroHH NO$_2$ profiles of the Sentinel-5P source pixel for Matimba on the 25[th] of July 2020 at 12:00 UTC. (b) Histogram of the default and recalculated AMFs of the TROPOMI pixel containing the Matimba power plant based on MicroHH NO$_2$ profiles

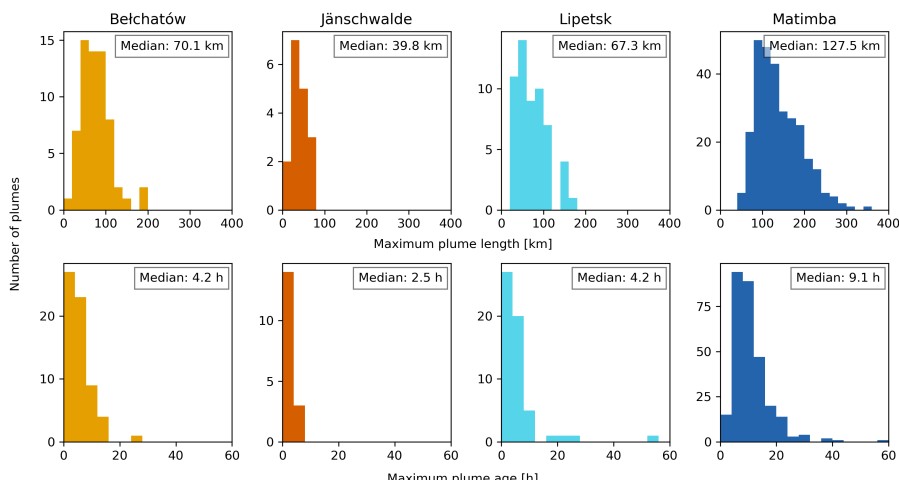

**Figure A6.** Maximum lengths and ages of detected plumes from TROPOMI observations for the years 2020 and 2021. Ages were calculated by dividing the plume length by the profile weighted wind speed at the source.



*Author contributions.* SM conducted the analysis and authored the manuscript with input from all co-authors; EK processed MicroHH model simulations into pseudo satellite images; MK provided insight in the MicroHH simulations and $NO_x$ chemistry; DB and AD offered constructive manuscript feedback; GK coordinated and supervised the project.

*Competing interests.* The authors declare no competing interests.

*Acknowledgements.* The research was funded by the Horizon Europe CORSO project (no. 101082194) with additional funding by the Swiss State Secretary for Education, Research and Innovation (SERI, no: 22.00422). We like to acknowledge the ICOS Carbon Portal for providing the computational resources needed for the analysis shown in this paper.



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
