# Peer review of "A light-weight $NO_2$ -to- $NO_x$ conversion model for quantifying $NO_x$ emissions of point sources from $NO_2$ satellite observations"

_EGUsphere, 2024_

## Referee Comment (RC2)

Review of "A light-weight NO2 to NOx conversion model for quantifying NOx emissions of point sources from NO2 satellite observations"

Sandro Meier, Erik F. M. Koene, Maarten Krol, Dominik Brunner, Alexander Damm, and Gerrit Kuhlmann

**General Comments**

This study investigates the method used to convert NO2 emissions derived from TROPOMI to NOx emissions. The current method tends to underestimate NOx emissions by overestimating the NOx decay time throughout the course of plume lifetime. The new model improves this representation by calculating a time-dependent NOx:NO2 ratio. The model is based on model simulations that consider detailed chemistry and meteorology over 4 NOx emission sites. This paper logically explains the authors' methods, reasoning, and results. This paper also does a good job of tracking error and explaining NO-NO2-O3 chemistry to explain the results. My concerns lie mostly in a lack of description of some background information, and the applicability of these results to greater time periods, regions, and weather conditions. I recommend publishing this paper after revisions by the authors, detailed below.

**Specific Comments**

1. More description of MicroHH and CSF would be very helpful in the Introduction. Information that I would find useful include inputs to each model, outputs from each model, general physical principles underlying each model, spatial and temporal resolutions, and the reason why each model was chosen for your project (i.e., the benefits of those models compared to similar model options). Some of these aspects are discussed in the Methods section, but it would be helpful to have more context for them when they're first introduced, especially for people who are familiar with TROPOMI and NOx chemistry, but not with this style of modeling.
2. I'm curious how important the NO:NO2 split was when modeling the NOx emissions in MicroHH (line 123). Were any sensitivity simulations performed by varying the split?
3. Line 134 states that only 2 days were selected to derive the fit parameters. I would suspect that the fit parameters are quite specific to those days (which is commented on in later sections). While it is shown that the new method improves NOx emission estimates compared to the existing method with a constant NOx:NO2 ratio, I wonder whether this conclusion holds when different days are used. What were some of the meteorological qualities of the chosen days, e.g., were wind speed, wind direction, humidity, etc. typical? How would the simulation respond to days with abnormal weather or emissions phenomena?
4. When introducing Equation 5, it could be helpful to explain why a negative exponential function was used, e.g., to match observations or to fit a first order reaction rate.
5. The first paragraph of Section 3.2 comments on the limited availability of cloud-free, plume-detected images. I'm concerned about N = 17 for Janschwalde in Figure A3. Can these numbers of images provide statistically-significant results? This topic is further discussed in Section 4.2, which is helpful, and I think additional commentary about why these sites were chosen – despite their lack of data and the overlapping plumes – would provide helpful context.

6. In Figure 7, please state what the relative mean bias error is relative to. I assume it's relative to the bottom-up emission estimate since Lipetsk has no data, but the legend does not make that clear. I originally thought that the large, filled bars were for the bottom-up NOx estimates because they do not have a black outline.

7. In the final paragraph of Section 3.2 (starting line 296), you comment on the fit parameters needing to be specific to each location. You comment on the error expected when applying this framework to other seasons or regions in Sections 4.1 and 4.2, which is helpful. Can you please comment on whether this framework is only intended to be used for local studies, and whether more research is being done to investigate its usage on global TROPOMI data?

**Technical Corrections**

1. For clarity when first introducing the topic of NO2-to-NOx conversion, please change the sentence beginning on line 51 to "To retrieve NOx emissions, it is therefore necessary to convert NO2 emissions to NOx emissions". As-written, someone new to the field may think that the conversion is chemical, and not analytical, especially following the sentence about the chemistry of the NOx family.

2. Figure A1 could benefit from a top axis showing distance as is done in Figure 4.

---

## Author Comment (AC1)

**Answer to the reviewers:**

**Reviewer 1**

AC: We like to thank Reviewer 1 for the valuable review. We have revised the manuscript accordingly and provide a point-by-point response to all comments below. Our replies are written in blue.

The manuscript presents a concrete model for the conversion of NO to NO2 in point source emission plumes - an effect that is well known in principle, but hard to quantify in concrete cases due to its dependency on multiple parameters. Consequently, this process has so far been ignored in satellite-based point source emission estimates.

The authors propose a simple parameterization for the NOx to NO2 ratio and provide concrete parameters for four selected point sources.

It is also shown that ignoring the in-plume conversion can result in significant underestimation of point source emissions from satellite data.

Thus, the topic of the study matches the scope of ACP, and the study contributes an important scientific progress.

The paper is generally well written and the presented material and the conclusions drawn are mostly comprehensible. In some cases, the authors should provide more quantitative or concrete information, as specified in the detailed comments below.

My main concern is that the paper gives concrete recommendations for an improved NO2 to NOx upscaling for 4 point sources which are based on only 2 days model simulation each.

The representativeness of the selected days for simulations is discussed rather qualitatively.

As the conversion time has to be expected to depend on stability, the wind speed probably plays an important role - an aspect that is not explicitly investigated or discussed in the current manuscript.

Thus, the authors should at least add one further model simulation and fit of f(t) for each location, such that for each location a high wind speed and a moderate wind speed case are investigated.

The discussion of the representativeness of the model simulations and the impact of stability/wind speed should be updated/added accordingly.

I thus recommend publication after (a) additional simulations have been added, (b) the discussion of the representativeness of the simulation results is updated accordingly and the impact of meteorological stability is discussed, and (c) the additional comments listed below have been accounted for.

We agree that additional simulations would be useful for providing additional insights into the variability and drivers of the $NO_2$ to $NO_x$ conversion. The simulations used in this study were conducted in the CoCO2 project with the aim to validate how well atmospheric models can simulate plumes with available airborne and satellite measurements. Since the computational costs of the simulations are very high (400-1000 node hours per case, Krol et al. 2024, in preprint: https://egusphere.copernicus.org/preprints/2024/egusphere-2023-

2519/), we currently do not have the resources for additional simulations. We are aware that this is a limitation of our study and already discuss this throughout the manuscript (e.g., Abstract, Section 3.2, Section 4 and 5).

To address the representativeness of wind speed in the MicroHH simulations, we have conducted an additional analysis of the wind speed in the MicroHH simulations and ERA-5 and added them to the supplement and the appendix (Fig. S1 and A7). For this, we have compared the vertically integrated wind speeds in MicroHH for the time steps 8 to 14 UTC for both simulated days with the vertically integrated wind speeds from ERA-5 at TROPOMI overpass time for the years 2020 and 2021. Supplementary Figure 1 shows that wind speeds in the MicroHH simulations cover a wide range (2 and 8 m/s in the period used for fitting f(t)). The median wind speed in the MicroHH simulations and the ERA-5 data are about 5 m/s, showing that the MicroHH simulations are representative for wind conditions where TROPOMI observations are available.

[Figure]

[Figure]

Supplementary Figure 1: Temporal evolution of vertically weighted wind speeds in the MicroHH simulations of Bełchatów, Jänschwalde, Lipetsk, and Matimba as a function of simulated time steps. Grey shading represents the time steps used in the analysis.

Appendix Figure 7: Vertically integrated ERA-5 wind speeds at TROPOMI overpass time for the years 2020 and 2021, and vertically integrated simulated wind speeds in MicroHH for the time steps 8 to 14 UTC on both simulated days. For both variables, the wind speed was sampled at the location of the sources Bełchatów, Jänschwalde, Lipetsk, and Matimba.

We have also conducted further analyses to assess the effect of wind speeds and PBL stability on the $NO_x:NO_2$ ratios. The left figure below shows the $NO_x:NO_2$ ratios of the MicroHH simulation of Matimba for time steps with high and low wind speeds. The $NO_x:NO_2$ ratios of two consecutive time steps with similar wind speeds are less similar than the ratios at the same time of the day. This shows that wind speeds are not the driving factor of the $NO_x:NO_2$ ratios, instead it is more likely that other factors such as the photolysis rates or background concentrations with their diurnal cycle exhibit a stronger influence.

The right figure depicts $NO_x:NO_2$ ratios for a stable PBL in the morning and a well-mixed PBL in the afternoon. For this, we have selected the time steps in the morning and afternoon of the same day with the most similar wind speeds and photolysis rates. The $NO_x:NO_2$ ratios of two consecutive time steps are similar but differ between morning and afternoon. It is likely that $O_3$ is entrained more efficiently into the plume in a well-mixed PBL, leading to lower $NO_x:NO_2$ ratios. However, as the NO and $NO_2$ background concentrations decrease over the course of the day (see Supplementary Figure 2 below), the difference in $NO_x:NO_2$ ratios cannot clearly be assigned to the effect of stability.

[Figure]

NO$_x$:NO$_2$ ratios of the MicroHH simulation of Matimba. Left: Time steps with high and low wind speeds. Right: Time steps with a stable PBL in the morning and a well-mixed PBL in the afternoon.

As described on Line 134, we analysed the time steps 8 to 14 UTC for both simulated days instead of only the ones at TROPOMI overpass time to derive more robust NO$_2$-to-NO$_x$ conversion factors that better represent varying atmospheric and site conditions. In this way, we incorporate different wind speeds, background concentrations and atmospheric stabilities (stable PBL in the morning and well mixed PBL in the afternoon) in our NO$_2$-to-NO$_x$ conversion parameters. The uncertainties provided for the parameters (m, r and f0) reflect these varying conditions.

**Additional comments:**

General: Ratios are denoted as "NO2-to-NOx", "NO2 to NOx", or (reciprocal) "NOx:NO2". Please be consistent throughout the manuscript.

Thanks for pointing this out! The reasoning behind the notations is the following: NO$_2$-to-NO$_x$ is used as a specification of the conversion factor (e.g. "NO$_2$-to-NO$_x$ conversion factor") whereas NO$_2$ to NOx denotes the conversion of one into the other (e.g. "Conversion of NO$_2$ to NOx line densities). "NO$_x$:NO$_2$" denotes the ratios of NO$_x$ to NO$_2$. We have updated the instances where this scheme was not followed.

Line 21: I don't see (yet) the step towards a "global uniform" application. The study demonstrated that conditions can be very different for the investigated point sources.

We changed "uniform" to "consistent" here and at other places (L. 35 and 435). We think that our approach of modelling f(t) as negative exponential function works well enough to be applied globally, but parameters (m, r and f0) indeed need to be adjusted to the specific conditions, which vary in space and time.

Line 42: I don't think that the NO2 data from satellite has a higher accuracy than for CO2. The challenge for CO2 is to accurately measure the (small) CO2 *excess* on top of the high background.

Yes, this is an imprecise formulation. We have adjusted it accordingly.

Line 52: Note that some recent studies do not use an "one for all" factor of 1.32, but account for the spatial variability of the ratio: e.g.

- Beirle et al., 2021 (already cited)

- Lange, K., Richter, A., and Burrows, J. P.: Variability of nitrogen oxide emission fluxes and lifetimes estimated from Sentinel-5P TROPOMI observations, Atmos. Chem. Phys., 22, 2745–2767, https://doi.org/10.5194/acp-22-2745-2022, 2022.

We have added "derived spatially varying conversion factors" in L54 and added the missing paper by Lange et al. 2022.

Fig. 2 (b): Please explain the black lines (current legend is not helpful). For positive distance, the black line seems to be a Gaussian fit (eq. 2?), but what is the meaning of the black line for the first subplot with negative distance, and what is done with these values?

The figure caption was adjusted. As the reviewer correctly assumed, for positive distances, a Gaussian curve as in Eq. 2 is fitted. For negative distances, the upstream polygon is divided into slices of 5 km width. For each of these sub-polygons, the VCDs are summed up. However, in the current study, this procedure is not important as the estimated value of the upstream polygon is not used. This has been clarified in the revised figure caption.

Line 101: Subscript "eff" should be in text mode.

Was adjusted.

Line 101: Please add a ref to section 4.3 here.

Was adjusted.

Lines 103-105: What time interval is used for the emission fit? Do the results depend on this choice?

Currently, the line densities along the entire detected plume are used to fit the emissions (fig 2c).

Line 114: Please provide further details about how the chemistry was "tuned". Was this done once for all or did you need to tune MicroHH for each location?

The MicroHH chemistry was "tuned" only once to be consistent with the IFS chemistry scheme. This is described in detail by Krol et al. (2024), which is available as preprint here: https://egusphere.copernicus.org/preprints/2024/egusphere-2023-2519/.

Line 121: Please provide further details and list the used bottom-up emissions with reference to table 1.

We have added the used emissions to Table 1. For details, we refer to Krol et al. 2024.

Section 2.1.2 / Table 1: Please explain how the simulation periods have been selected - probably because TROPOMI indicated a clear NO2 plume?

Yes, the simulated dates were selected based on the availability of cloud-free TROPOMI images as well as flight campaigns. An explanation was added.

As written above, I would encourage the authors to add additional cases for each location. Ideally one day with high vs. a day with moderate wind speed. In any case, u_eff should be added to Table 1, as this is an important characteristic for the plume properties.

We agree that having more simulations covering a wider range of meteorological and background conditions would be insightful. However, additional simulations are computationally very expensive and we currently do not have the resources for this.

Since the wind speed varies strongly during simulations, we added a figure showing the evolution of $U_{eff}$ to the supplement showing that our simulations already include a day with moderate and high wind speed (Fig A7).

Note that we specifically use time-since-emissions in our approach to minimize the dependency on the effective wind speed on the $NO_2$ to $NO_x$ conversion. For example, the figure below shows the $NO_x:NO_2$ ratios as a function of distance from the source (left) and as a function of time since emission (right). One can see that especially close to the source, the spread of $NO_x:NO_2$ ratios is smaller in the figure on the right.

[Figure]

$NO_x:NO_2$ ratios of the MicroHH simulation of the time steps 8 to 14 UTC of both simulated days as a function of distance from the source (left) and as a function of time since emissions (right).

Section 2.2.1: It is mentioned later (3.2), but it would have helped me understanding the data processing if a sentence like "TROPOMI data were selected where plume detection worked successfully (see section 3.2 and Fig. A3)" would appear somewhere in section 2.2.1.

A corresponding sentence was added.

Line 164: I don't understand this point: why should a variable (VCD dependend) precision be problematic for the calculation of line densities?

In the TROPOMI NO2 retrieval, there are additive terms (stratospheric correction) and multiplicative terms (AMF). Due to the latter the NO2 VCD precision is inevitably correlated with the VCD and can be substantially larger than 1e15 close to strong emitters.

This has a direct impact on the uncertainty of the emission estimate: If the AMF is wrong by 20%, the derived emissions are off by 20% as well (if the rest of the algorithm is perfect).

Thus, for a meaningful error propagation, the individual (VCD dependend) precision values have to be taken into account.

To determine the line density, we fit a Gaussian curve (Eq. 2) to the observed $NO_2$ VCDs using a weighted least squares (WLS) method. Since the errors in $NO_2$ VCD depend on the retrieved VCDs (due to multiplicative terms in the error calculation), computing the weights from the reported errors would give lower weight to higher VCDs and, as a result, would tend to underestimate the line density. In order to avoid this, we set the uncertainty (and hence the weights) to a constant value. We have changed the sentences for clarification.

Line 181: The "mixture of both" (mixture of correct concentration profile of plume and background?) is confusing. Please reword.

We removed the confusing part.

Line 188: This effect would mean that the observed SCD increases with distance, and thus (if not properly accounted for in the AMFs) the VCD as well. I.e. this effect interferes with the NO to NO2 conversion. Please give an estimate on the AMF change due to profile changes within the plume transport and discuss how far this could affect the estimated conversion rate r.

Since the AMF correction is only applied to TROPOMI data and not to MicroHH data, there is no influence of the AMF correction on the estimated conversion rate r.

However, the replacement of PBL $NO_2$ profiles is important as shown in Figure A2. To study the error introduced by the assumption of constant $NO_2$ profiles within the PBL, we have conducted further analyses using the MicroHH simulations. First, we calculated the expected SCDs for the MicroHH simulation. The SCDs were then converted back to VCDs using AMFs which were calculated under the assumption of a constant $NO_2$ profile within the PBL. From the resulting VCDs, we estimated the $NO_x$ emissions and compared them to the estimated emissions using the true MicroHH VCDs. The figure below shows that assuming constant $NO_2$ in the PBL results in an emission underestimation of about 8.5%. We added the following sentence to the manuscript:

*"We conducted a sensitivity study with the MicroHH profiles of Matimba to study the impact of varying $NO_2$ profiles inside the plume, showing that the use of the AMF calculated when assuming constant $NO_2$ PBL concentrations to convert SCDs to VCDs leads to an underestimation of about 8.5% compared to when using the true MicroHH $NO_2$ VCDs (see Figure S5 in the supplement)"*

[Figure]

Supplementary Figure 5: Relative difference in estimated NOx emissions when re-calculating MicroHH VCDs using the updated AMFs where the NO2 mole fraction was set to 5*10^9 mol/mol within the PBL of the detected plumes and estimated NOx emissions when using the true simulated VCDs in MicroHH.

Line 238: Please provide more quantitative information here and name the main reasons for the large differences:

- assumed emissions should be added to Table 1 and can be refered here

Was added.

- "meteorological conditions" are basically u_eff which should be added to Table 1 as well

It is not only u_eff (which is used to convert line densities into fluxes) but also turbulence (which determines the mixing in of surrounding air masses which accelerates the oxidation of NO to $NO_2$) and temperature (which determines the reaction rates).

- while solar irradiance is of course important in general, I would not understand if this would explain the observed differences here, as for all cases cloud free conditions around noon are considered.

Yes, all simulations are cloud free but have different photolysis rates due to different latitudes.

But if you find this to be actually different, please list the numbers as well.

Was added.

- please specify the assumed O3 concentrations for the four point sources.

We have added a figure showing the vertically averaged background concentrations of the lowest kilometre to the supplement (Fig S2).

[Figure]

Supplementary Figure 2: Temporal evolution of mean background concentration of reactive trace gases in the lowest kilometer for the MicroHH simulations of Bełchatów, Jänschwalde, Lipetsk, and Matimba. Grey shading represents the time steps used in the analysis.

- are the VOC concentrations very different for the four locations, and does this explain the far higher values for Matimba?

They do differ but are likely not the main reason for the differences in $NO_x:NO_2$ ratios. For example, the high NOx emissions in combinations with the lower $O_3$ background concentrations causes $O_3$ to remain fully titrated longer at Matimba than at other places.

Section 3.2: The authors compare their derived f(t) with the often-used value of 1.32. For Jänschwalde or Lipetsk, f0 is close to 1.32, and the remaining discrepancy is the effect of NO to NO2 conversion in the plume.

But for Belchatow and particularly for Matimba, there are two reasons for the differences using 1.32: in addition to the ignored change of the in-plume NOx/NO2, also the final ratio is quite different.

Indeed, analysing the $NO_x:NO_2$ ratio from CAMS data for the two simulated days for each source shows that they are between 1.2 and 1.5 for the time steps used in this study (see Fig Supplementary Figure 3 below). We have adjusted the following sentence in Section 3.1 and added reference to this figure.

"*Furthermore, the background concentrations of $O_3$ and VOCs that are different for all simulations have an strong influence on the $NO_x:NO_2$ ratios and partly explain the higher values of f0 for Bełchatów and Matimba (see Figures S2, S3 and S4 in the supplement)*"

[Figure]

Supplementary Figure 3: NO$_x$:NO$_2$ ratios based on background NO and NO$_2$ concentrations from CAMS (for details see Krol et al. 2024). Grey shading represents the time steps used in the analysis.

As mentioned above, some recent studies account for the general place-dependent value of the NOx/NO2 ratio. In order to separate the impact of the wrong "background" value from the ignored in-plume NO to NO2 conversion, it would be very interesting to see comparisons of f(t) with the (still constant, but locally adjusted) value of f0, in addition or instead of 1.32.

As seen in the figure above, the background NO$_x$:NO$_2$ ratios from CAMS are already higher than 1.32. Thus, the use of these values to convert NO$_2$ to NO$_x$ line densities instead of 1.32 would increase the estimated emissions by 10-20%. However, as seen in the blue curve in Fig 1, the NO$_2$ concentration (or line densities) increases after the emission and decreases afterwards. If we would scale it with different values for f0 to obtain NOx line densities, it would still maintain its shape. Consequently, fitting an exponential decay function to these NOx line densities (Eq. 4) would not work very well and result in a still strongly overestimated NOx lifetime. We added a sentence why a negative exponential function is needed in section 2.1.3:

*"A negative exponential function was chosen for the conversion of NO$_2$ to NO$_x$ to account for the initially high ratios at source that decrease with time due to the oxidation of NO."*

Figures 6 and 7: shaded bars have different meanings in both figures - this should be made consistent.

We have adjusted the figure and legend accordingly.

Lines 300-302: Please comment on the needed effort for running the required simulations and the potentially required local "tuning".

The computational cost of running a simulations as presented in this study ranges between 400 to 1000 node hours per case (Krol et al. 2024, in preprint: https://egusphere.copernicus.org/preprints/2024/egusphere-2023-2519/). On top of that, the pre- and post-processing requires further working hours.

As written above, the MicroHH chemistry was "tuned" only once to be consistent with the IFS chemistry scheme (for details, see Krol et al. 2024). No tuning would be required for individual simulations. We have added a sentence to explain the used of these simulations

Section 4.1: The selection of TROPOMI overpasses where a clear downwind plume shows up implies stable wind conditions. I would expect that the fitted f(t) is thus representing this special case, and average NO to NO2 conversion is probably quicker for less stable conditions.

This "selection bias" should be discussed somewhere.

We argue that there is no selection bias with regard to the method developed in this study because the MicroHH simulations represent cloud-free turbulent conditions with well-mixed PBLs. As only cloud-free TROPOMI images can be used to estimate emissions, the simulated conditions are representative for the prevailing conditions of successful TROPOMI retrievals.

Line 308: Again, I don't see photolysis as critical component here, as the selection of cloud free observations during noon results in high photolysis rates for all considered sites. If I am wrong please provide some quantitative statements on this.

Indeed, all simulations are cloud free but have slightly different photolysis frequencies due to different latitudes. We have added a figure to the supplement showing the photolysis frequencies used for the simulations (Fig S4). Additionally, we have removed photolysis from the sentence to make it clearer that it is not a driving parameter for differences in the derived parameters for the $NO_2$-to-$NO_x$ conversion.

[Figure]

Supplementary Figure 4: Temporal evolution of solar zenith angle and frequencies of simulated photolysis processes in the MicroHH simulations of Bełchatów, Jänschwalde, Lipetsk, and Matimba. Grey shading represents the time steps used in the analysis.

Line 309: "Trace gas" would be O3, NOx and VOC, or something else as well?

Yes, all gases which are relevant for NOx chemistry (O3, NOx, VOCs, $CH_4$, CO etc.). This is not further specified because it was briefly touched on in the introduction. We reworded this sentence to avoid confusion.

Line 319: The authors should at least provide one more simulation per location with different conditions (wind speed) in order to better assess the question of representativity.

As mentioned above, running further simulations would be very insightful but are currently not planned due to the high computational costs.

We argue that the use of the time steps 8 to 14 UTC for both simulated days to derive more robust $NO_2$-to-$NO_x$ conversion factors that represent varying atmospheric and site conditions. In this way, we incorporate different wind speeds and atmospheric stabilities.

Despite the small number of simulations, the current analysis was able to show that the use of a time-since-emission dependent conversion factor for $NO_2$ to $NO_x$ can significantly decrease the bias in estimated emissions.

Lines 365-366: Please discuss how much of the discrepancy is due to ignorance of in-plume conversion, and how much is due to wrong "background" pss (which can easily be avoided) by the comparison of f(t) with f0, see comment to sect. 3.2.

See comments above

Section 4.3

Taking u_eff just from the source location is a rough simplification. In 4.3, the authors discuss the difficulties of calculating appropriate effective wind speeds downwind due to plume mixing.

Apart from this, also the wind speed at a typical altitude might just change downwind. I would expect this to be rather common. As far as I understand, this effect is currently ignored completely.

Please add a discussion of this effect and its impact on the conversion of distance to time and the applied fits.

We have analysed ERA-5 wind fields around the source and found that for our cases temporal variability was more important the spatial variability. This issue was addressed in an analysis not shown in this paper, where we integrated the wind backward in time to account for the wind history. However, the results showed that it had no significant influence on the estimated emissions. As the issue of the representativeness of wind speeds to convert line densities into fluxes is not inherent to the topic discussed in this paper but to all methods used to estimate emissions from remote sensing data, we decided to not further discuss this in this paper. We also found that using time-since-emissions instead of along-plume distances reduces the spread of $NO_x:NO_2$ ratios.

Line 393: Note that in Beirle et al. (2019), the correction factors of 1.35 and 1.98 are not intended to correct the total (tropospheric) VCD, but the VCD *excess* only (the enhancement of the VCD caused by the local point source), as the divergence of the flux automatically removes the local background. This is discussed in more depth in section 3.3 in Beirle, S., Borger, C., Jost, A., and Wagner, T.: Improved catalog of NOx point source emissions (version 2), Earth Syst. Sci. Data, 15, 3051–3073, https://doi.org/10.5194/essd-15-3051-2023, 2023.

Was adjusted

Line 423: I would expect that stability / wind speed is a key component in this list.

Was added

Fig. A1:

- I would expect that conditions are quite different during night. Due to lack of photolysis, f0 should approach 1. I would propose to skip (or at least to mark) nighttime in this plot.

Please note that the time is given in UTC whereas the local time is UTC+2 for all sources. It is indeed the case that the $NO_x:NO_2$ ratio approaches 1 towards the end of the plume at night. We have adjusted the figure to show daytime hours as well as the time steps used for our analysis.

[Figure]

Appendix Figure 1: $NO_x$:$NO_2$ ratios for 48 individual hourly time steps of the MicroHH simulations of Bełchatów, Jänschwalde, Lipetsk, and Matimba as a function of time since emission, highlighting the spatiotemporal patterns of the $NO_x$ chemistry. Yellow shading represents daytime and grey shading represents the time steps used in the analysis.

- Why is there a gap of several hours for Lipetsk?

The plume was only partially within the domain, due to low and different wind speed, such that the center line could not be fit to the plume which is needed for computing time-since-emissions.

- For Matimba, time 6-12 UTC, the NOx/NO2 seem to *increase* with time - how can this be???

The reason for these increases is the fact that the Matimba plumes are very long (see Figure A6). Thus, parts towards the end of the plume were emitted several hours earlier into the free troposphere

These local increases towards the end of the plume are accounted for when fitting the $NO_2$-to-$NO_x$ conversion factors as the increases lead to higher uncertainties in estimated line densities. These uncertainties are propagated when fitting the $NO_2$-to-$NO_x$ conversion factors.

**Reviewer 2**

AC: We like to thank Reviewer 2 for their helpful comments. We have revised the manuscript based on their suggestions and provide a point-by-point response below. Our replies are written in blue.

**General Comments**

This study investigates the method used to convert NO2 emissions derived from TROPOMI to NOx emissions. The current method tends to underestimate NOx emissions by overestimating the NOx decay time throughout the course of plume lifetime. The new model improves this representation by calculating a time-dependent NOx:NO2 ratio. The model is based on model simulations that consider detailed chemistry and meteorology over 4 NOx emission sites. This paper logically explains the authors' methods, reasoning, and results. This paper also does a good job of tracking error and explaining NO-NO2-O3 chemistry to explain the results. My concerns lie mostly in a lack of description of some background information, and the applicability of these results to greater time periods, regions, and weather conditions. I recommend publishing this paper after revisions by the authors, detailed below.

**Specific Comments**

1. More description of MicroHH and CSF would be very helpful in the Introduction. Information that I would find useful include inputs to each model, outputs from each model, general physical principles underlying each model, spatial and temporal resolutions, and the reason why each model was chosen for your project (i.e., the benefits of those models compared to similar model options). Some of these aspects are discussed in the Methods section, but it would be helpful to have more context for them when they're first introduced, especially for people who are familiar with TROPOMI and NOx chemistry, but not with this style of modeling.

   We like to keep the description MicroHH and the CSF in the introduction brief. However, we have added a few sentences to better introduce MicroHH and the CSF in the method section:

   *"The CSF method is a common mass-balance approach, which can be used to estimate emissions of point sources. An implementation of the approach is available in the open-source Python library for data-driven emission quantification (ddeq; \citet{Kuhlmann2023}). Since the CSF method divides a plume into several cross-sections perpendicular to the plume direction and establishes a plume-following coordinate system with along-plume and across-plume coordinates, it is ideal for studying the progress of the $NO_x$ chemistry inside of the plume."*

   We also have added some references to the respective subsections. More information about MicroHH can be found in the publication by Krol et al. 2024 (https://egusphere.copernicus.org/preprints/2024/egusphere-2023-2519/). The CSF method and its implementation of the ddeq library is described in more details by Kuhlmann et al. 2024 (https://doi.org/10.5194/egusphere-2023-2936)

2. I'm curious how important the NO:NO2 split was when modeling the NOx emissions in MicroHH (line 123). Were any sensitivity simulations performed by varying the split?

   We agree that this would indeed have been interesting to analyse and could influence the fitted $NO_2$-to-$NO_x$ conversion factors, but such sensitivity simulations are currently not available.

3. Line 134 states that only 2 days were selected to derive the fit parameters. I would suspect that the fit parameters are quite specific to those days (which is commented on in later sections). While it is shown that the new method improves NOx emission estimates compared to the existing method with a constant NOx:NO2 ratio, I wonder whether this conclusion holds when different days are used. What were some of the meteorological qualities of the chosen days, e.g., were wind speed, wind direction, humidity, etc. typical? How would the simulation respond to days with abnormal weather or emissions phenomena?

We agree that additional simulations would be useful to provide additional insights in the $NO_2$ to NOx conversion. The simulation used are part of the library of plumes generated in the CoCO2 project. Since the simulations computationally expensive, additional simulations are currently not possible. The wind speed in the MicroHH simulations cover a wide range (2 and 8 m/s in the period used for fitting f(t)) (see Fig. S1). The median wind speed in the MicroHH simulations and the ERA-5 data agree quite well (Fig A7). See also our reply Reviewer #1. Under extreme (weather) conditions, a method based on a short simulation period will have large uncertainties. However, days with abnormal weather events or conditions which strongly deviate from the simulations like no wind or strong wind speeds during storms would not be useful for emission quantification anyway (e.g. due to cloud cover and absence of valid satellite data, fast dilution of the plume etc.).

4. When introducing Equation 5, it could be helpful to explain why a negative exponential function was used, e.g., to match observations or to fit a first order reaction rate.

Thanks for pointing this out. We added a respective sentence.

5. The first paragraph of Section 3.2 comments on the limited availability of cloud-free, plume detected images. I'm concerned about N = 17 for Janschwalde in Figure A3. Can these numbers of images provide statistically-significant results? This topic is further discussed in Section 4.2, which is helpful, and I think additional commentary about why these sites were chosen – despite their lack of data and the overlapping plumes – would provide helpful context.

We estimate the temporal sampling bias as shown in Eq. 8, which result in total uncertainty of 20.5% for Jänschwalde (N=17, Table 4). The NOx biases due to the conversion factor is larger (~40%).

The MicroHH simulations were conducted as part of the CoCO2 project with the aim to assess how well different models could simulate emission plumes, considering availability of aircraft and satellite measurements for model evaluation. The simulated sources were not chosen to maximize the number of suitable TROPOMI observations per year.

6. In Figure 7, please state what the relative mean bias error is relative to. I assume it's relative to the bottom-up emission estimate since Lipetsk has no data, but the legend does not make that clear. I originally thought that the large, filled bars were for the bottom-up NOx estimates because they do not have a black outline.

Yes, the MBE is indeed relative to the bottom-up reported emissions. We have added this information to the figure caption. We have also adjusted the legend to increase comprehensibility.

7. In the final paragraph of Section 3.2 (starting line 296), you comment on the fit parameters needing to be specific to each location. You comment on the error expected when applying this framework to other seasons or regions in Sections 4.1 and 4.2, which is helpful. Can you please comment on whether this framework is only intended to be used for local studies, and whether more research is being done to investigate its usage on global TROPOMI data?

We think that the general framework (i.e. approximating the conversion by Eq. 5) can be applied globally, but more research is needed (and will hopefully done by us and others) to determine the variability of parameters m, r and f0 and their uncertainties.

**Technical Corrections**

1. For clarity when first introducing the topic of NO2-to-NOx conversion, please change the sentence beginning on line 51 to "To retrieve NOx emissions, it is therefore necessary to convert NO2 emissions to NOx emissions". As-written, someone new to the field may think that the conversion is chemical, and not analytical, especially following the sentence about the chemistry of the NOx family.

Thanks for pointing this out. We have reworded this sentence.

2. Figure A1 could benefit from a top axis showing distance as is done in Figure 4.

The time steps used in figure 4 have similar wind speeds and the calculation of an overall mean is reasonable. This is not the case for data over 48 hours because the temporal variations in wind speeds are very large (factor 2-3x larger at night than during daytime). As a result, calculating an overall mean wind speed to add a length scale would not be very meaningful.